# Dipole field in nitrogen-enriched carbon nitride with external forces to boost the artificial photosynthesis of hydrogen peroxide

Zhi Li[1], Yuanyi Zhou[1], Yingtang Zhou [2], Kai Wang [3], Yang Yun[4], Shanyong Chen [1], Wentao Jiao[5] ✉, Li Chen[6], Bo Zou [3] & Mingshan Zhu [1] ✉

Artificial photosynthesis is a promising strategy for efficient hydrogen peroxide production, but the poor directional charge transfer from bulk to active sites restricts the overall photocatalytic efficiency. To address this, a new process of dipole field-driven spontaneous polarization in nitrogen-rich triazole-based carbon nitride ($C_3N_5$) to harness photogenerated charge kinetics for hydrogen peroxide production is constructed. Here, $C_3N_5$ achieves a hydrogen peroxide photosynthesis rate of 3809.5 $\mu$mol $g^{-1}$ $h^{-1}$ and a $2e^-$ transfer selectivity of 92% under simulated sunlight and ultrasonic forces. This high performance is attributed to the introduction of rich nitrogen active sites of the triazole ring in $C_3N_5$, which brings a dipole field. This dipole field induces a spontaneous polarization field to accelerate a rapid directional electron transfer process to nitrogen active sites and therefore induces Pauling-type adsorption of oxygen through an indirect $2e^-$ transfer pathway to form hydrogen peroxide. This innovative concept using a dipole field to harness the migration and transport of photogenerated carriers provides a new route to improve photosynthesis efficiency via structural engineering.

Hydrogen peroxide ($H_2O_2$), one of the most important green chemicals, is widely utilized in industrial and environmental applications[1,2]. To overcome the present high energy consumption of industrial synthesis methods (viz. the anthraquinone oxidation process), artificial photosynthesis of $H_2O_2$ through oxygen and water on semiconductor photocatalyst surfaces has been extensively developed because it is an environmentally friendly, low-energy and safe process[3,4]. Therein, the rate-determining step composed of the $2e^-$ oxygen reduction reaction (ORR) or water oxidation reaction (WOR) depends on the photogenerated charge separation efficiency.

However, random charge flow causing rapid charge recombination behavior (bulk recombination (BR) and surface recombination (SR)) always reduces the efficiency of $H_2O_2$ photosynthesis (Fig. 1a)[5–7]. Compared with SR, which takes tens of nanoseconds, rapid BR usually occurs in picoseconds[8]. Therefore, reducing BR is vital to improve the separation efficiency of photogenerated carriers.

Structural engineering-induced dipole field effects to boost directional electron transfer provide very promising approaches to address the above challenges[9–11]. A dipole is defined as a pair of opposite charges "$q$" and "$-q$" separated by a distance "d". The

[1]Guangdong Key Laboratory of Environmental Pollution and Health, School of Environment, Jinan University, 511443 Guangzhou, China. [2]Marine Science and Technology College, Zhejiang Ocean University, 316004 Zhoushan, China. [3]State Key Laboratory of Superhard Materials, College of Physics, Jilin University, 130012 Changchun, China. [4]College of Environment and Resource, Research Center of Environment and Health, Shanxi University, 030006 Taiyuan, China. [5]Research Center for Eco-Environmental Sciences, Chinese Academy Sciences, 100085 Beijing, China. [6]Department of General Practice, First Medical Center, Chinese PLA General Hospital, 100853 Beijing, China. ✉e-mail: wtjiao@rcees.ac.cn; zhumingshan@jnu.edu.cn

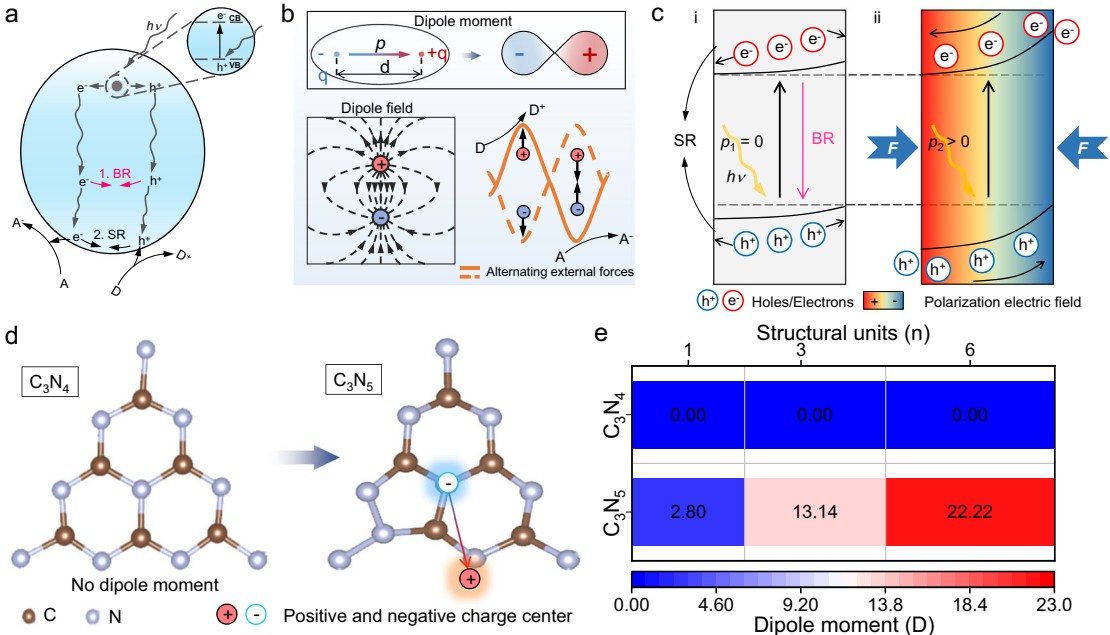

**Fig. 1 | Concept of a dipole field promoting photocatalytic carrier bulk separation. a** Photocatalytic carrier recombination processes including BR and SR. **b** Dipole moment and its electron cloud distribution, and dipole field and its change with external forces. **c** Mechanism of the field effect to promote the separation of photocatalytic carriers. **i** Photocatalyst without $p$ ($p_1 = 0$); **ii** Photocatalyst with $p$ ($p_2 > 0$) under an external force ($F$). **d** Structural unit and dipole moments of $C_3N_4$ and $C_3N_5$ with positive and negative charge centers. **e** Dipole moments of $C_3N_4$ and $C_3N_5$ with different structural unit numbers. D and A: Donor and acceptor molecule. $D^+$ and $A^-$: Donor cation and acceptor anion.

direction of the dipole moment ($p$) in space is from negative charge "$-q$" to positive charge "$q$"[12]. This dipole moment with an electron cloud distribution forms a dipole field, and the dipole field direction can be regulated by an alternating external force, which can be used to boost redox reactions (Fig. 1b)[13]. Specifically, this boosting is attributed to external force-induced internal polarization, which strengthens directional charge separation for redox. As shown in Fig. 1c, in the case of light only, for a photocatalyst without a dipole moment ($p_1 = 0$), photogenerated carriers (electrons and holes) exist in a random distribution state with SR and BR (Fig. 1c-i). However, when a photocatalyst has a dipole moment ($p_2 > 0$) with asymmetry of the unit cell structure, an internal spontaneous polarization field easily occurs in the internal bulk phase. This internal spontaneous polarization field leads to directional bulk phase photogenerated charge separation, finally facilitating the migration of charges to improve the redox potential under external forces (Fig. 1c-ii). Hence, structural engineering to form a dipole field is a new strategy to resolve the low efficiency of bulk charge migration and directional transfer in heterogeneous photocatalysts compared with traditional bulk phase heterojunctions. However, the relationship between dipole field and bulk charge migration has not yet been proven.

Carbon nitride ($C_3N_4$), with a suitable band position and excellent photocatalytic redox performance, has been the most commonly used material to produce $H_2O_2$[14]. Recent reports indicate that $C_3N_4$ has an in-plane piezoelectricity in the tri-s-triazine plane owing to the uniformly distributed triangular nanopores and local asymmetric structures[15,16]. However, density functional theory (DFT) reveals that the ideal single unit of $C_3N_4$ (Fig. 1d and Supplementary Fig. S1) does not show any dipole moment. This result demonstrates that $C_3N_4$ is not the ideal piezoelectric material to promote charge separation in the photosynthesis of $H_2O_2$. As shown in Fig. 1d, polymerization of the triazole and triazine framework to form a nitrogen-rich carbon nitride (viz. $C_3N_5$) leads to asymmetry of the structure, and a dipole moment is generated by the noncoincidence of the positively and negatively charged centers with a value of 2.80 D for a single unit. When the

number of units increases to 6, the dipole moment is enhanced to 22.22 D (Fig. 1e). This strong dipole moment means that dipole field-driven spontaneous polarization in $C_3N_5$ can be used to harness photogenerated charge separation kinetics. Here, we investigate the $H_2O_2$ photosynthesis efficiency of $C_3N_5$ with ultrasonic force vs. that of $C_3N_4$ via the difference in the dipole moment. The results show that the $H_2O_2$ production rate of $C_3N_5$ is 3809.5 μmol g$^{-1}$ h$^{-1}$, which exceeds most photosynthetic processes based on carbon nitride or piezo-photocatalytic processes. Combining theoretical calculations with in situ spectroscopic measurements, efficient directional charge separation from bulk to N active sites is found. Then, $O_2$ is adsorbed in a Pauling-type manner at the N active sites in $C_3N_5$ and undergoes an indirect 2 electron ORR from $O_2$ to $H_2O_2$ with intermediates of $^*O_2^-$ and $^*OOH$[3]. This simple-to-implement method fills an important gap for the bulk charge migration of photogenerated carriers via a structural engineering-induced dipole field and offers a brand-new understanding of the mechanism of photocatalytic $H_2O_2$ production.

## Results

### Structural characterization

$C_3N_4$ and $C_3N_5$ were prepared by one-step thermal polymerization, and the sheet-like morphologies are shown in transmission electron microscopy (TEM) images (Supplementary Fig. S2). The X-ray diffraction (XRD) pattern shows two diffraction peaks centered at 13.5° and 27.9° that can be attributed to the (100) and (002) planes of $C_3N_5$, respectively (Supplementary Fig. S3), corresponding to the in-plane structural ordering and interlayer stacking peaks of aromatic systems in graphitic materials[17,18]. Compared to $C_3N_4$, these diffraction peaks exhibit small shifts to large angles because of the reduction in the interlayer space.

The chemical bond features of $C_3N_5$ composed of a triazole and two triazine groups were explored by Fourier transform infrared spectroscopy (FT-IR) (Supplementary Fig. S4). Both $C_3N_4$ and $C_3N_5$ show sharp peaks at 810 and 891 cm$^{-1}$, corresponding to the condensed C-N heterocycles of the triazine moiety, and the peaks at

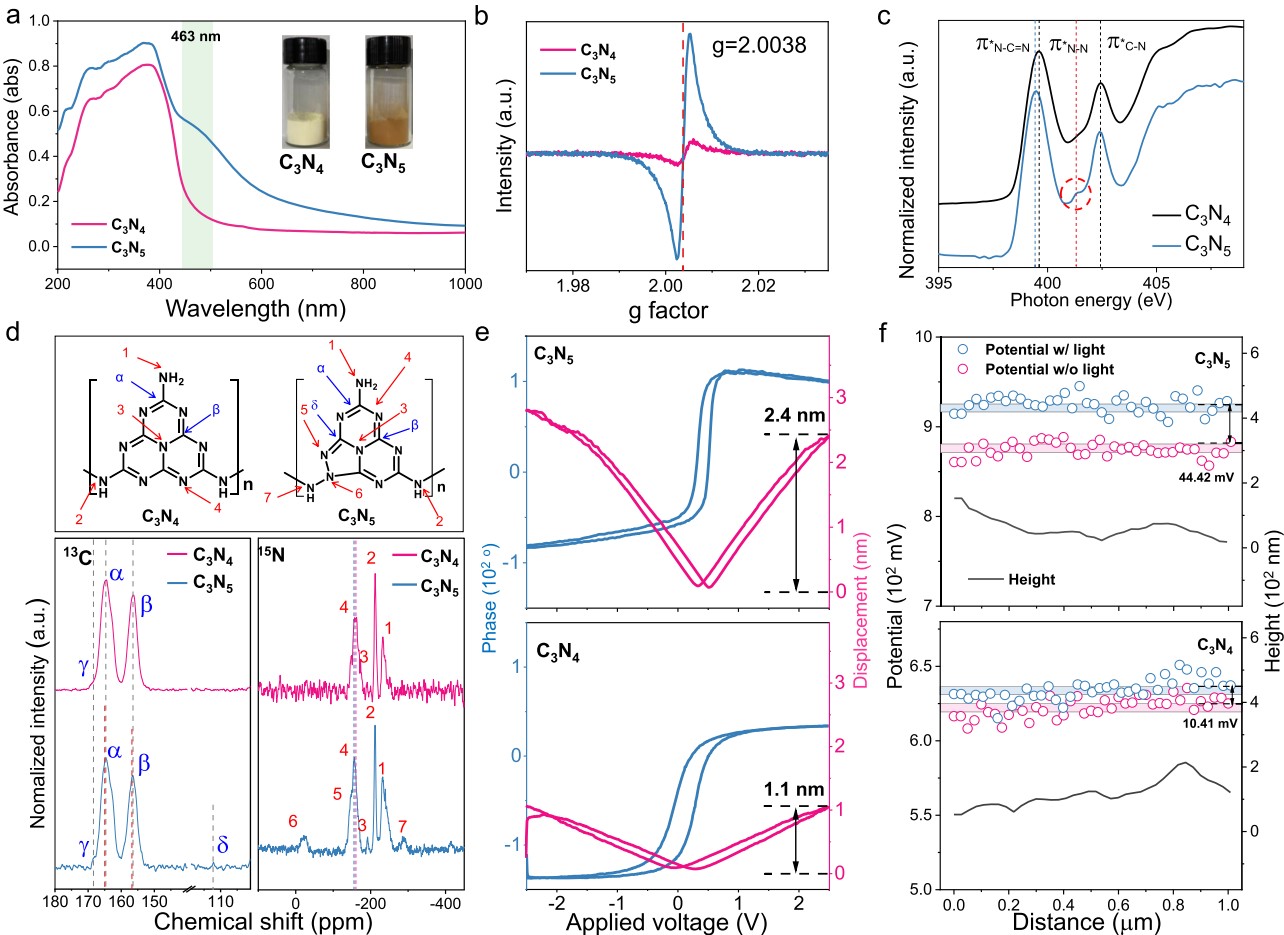

**Fig. 2 | Structural characterization of C₃N₄ and C₃N₅. a** UV–Vis absorption spectra. The inset shows a photograph of the two samples. **b** EPR spectra in terms of the g-factor. **c** N K-edge NEXAFS spectra. **d** Structural model representation and solid-state ¹³C and ¹⁵N NMR spectra. **e** Phase hysteresis loop and amplitude butterfly loop of C₃N₅ and C₃N₄. **f** Surface potential from KPFM images of C₃N₄ and C₃N₅ with (w/) and without (w/o) light irradiation.

1200–1700 cm⁻¹, including those at 1458 and 1635 cm⁻¹, are attributed to the stretching modes of C-N heterocycles of triazine[19]. Notably, triazole-based C₃N₅ displays intense peaks at 740 and 775 cm⁻¹, revealing the existence of a heterocyclic N-N bond of the triazole moiety, while no corresponding peaks are found for C₃N₄[20]. Another new peak at 2180 cm⁻¹ for C₃N₅ is assigned to cyano groups (-C≡N) converted from the terminal -C-NH₂ in the melon structural unit. All samples show a broad band at 3200–3400 cm⁻¹ owing to H₂O in the material and the terminal amino group of the CN framework. UV–visible absorption spectroscopy was used to explore the optical absorption properties. Compared to C₃N₄, C₃N₅ shows an absorption edge at approximately 463 nm (Fig. 2a) due to the n–π* electronic transitions of the conjugated CN framework[21]. Moreover, C₃N₅ displays an absorption edge redshift to 675 nm, which is attributed to the introduction of triazole groups inducing n–π* electronic transitions in the conjugated heterocyclic ring systems and π–π* transitions from sp² hybridization of C and N in the triazole clusters[22]. Electron paramagnetic resonance (EPR) spectroscopy was also used to explore unpaired electrons from structural detects (Fig. 2b). Compared to C₃N₄, C₃N₅ shows a higher single Lorentzian line at g = 2.0038, corresponding to the delocalized electrons on the heptazine rings[23], confirming that the introduction of the triazole ring facilitates the electron transport of carbon nitride.

Near-edge X-ray absorption finestructure (NEXAFS) analysis was used to explore the chemical bonds of triazole-based C₃N₅. The C K-edge NEXAFS spectra (Supplementary Fig. S5) of C₃N₅ and C₃N₄ show characteristic excitations, including 1 s→π*_out of plane C=C at

-285.2 eV and 1 s→π*_N-C=N at 288.0 eV[24]. Compared with C₃N₄, C₃N₅ displays a blue shift of 1 s→π*_N-C=N excitation, which is attributed to the increased N-C = N bond strength upon the formation of triazole moiety. The N K-edge NEXAFS spectra (Fig. 2c) of two samples show similar response at 399.3 and 402.5 eV, attributing to the 1 s→π*_N-C=N and π*_C-N resonance. Note that, relative to C₃N₄, triazole-based C₃N₅ displays a redshift of 1 s→π*_N-C=N excitation and a new peak at 401.1 eV, which are attributed to the formation of triazole moiety and 1 s→ π*_heterocyclic N-N of triazole, respectively[24,25]. Solid-state nuclear magnetic resonance (NMR) spectroscopy was further used to explore the chemical structure (Fig. 2d). The ¹³C NMR spectra of both samples display two peaks at 165 and 156 ppm and a weak peak at 169 ppm, attributed to C_{2N-NHx} (α) and C_{3N} (β) in the heptazine units in C₃N₅ and C_{-NH-C≡N} (γ), respectively[26,27]. Notably, a new peak at 111.9 ppm is observed for C₃N₅, corresponding to C_{HN-C(H)=N} (δ) in the triazole group[22], and all peaks of C₃N₅ are slightly shifted relative to C₃N₄ due to the triazole group in C₃N₅. The ¹⁵N NMR spectrum of C₃N₄ displays four signals at −156.6, −191.6, −211.6 and −227.6 ppm, assigned to NC₂ (N4), central NC₃ (N3), bridged NH (N2) and NH₂ (N1), respectively[27]. Similar to C₃N₄, the ¹⁵N NMR spectrum of C₃N₅ exhibits similar peaks, and new signals at −22.4, −146.7 and −288.8 ppm are observed. These new signals are attributed to C-N (N6)-N₂, C = N (N5)-N and N-NH (N7) from the triazole group in C₃N₅, respectively, due to the strong nitrogen-proton coupling with neighboring ammonia groups[28,29]. The NMR signals of C (α), C (β) and N (4) display a slight shift to low fields owing to the reduced shielding effect of electron-withdrawing groups (neighboring C or N groups), which is derived from the extra N atoms in the triazole

ring[27]. These experimental findings provide strong evidence of the coexistence of triazole and triazine moieties in the tetrazole-derived $C_3N_5$ materials, which is further confirmed through X-ray photoelectron spectroscopy, organic elemental analysis, matrix-assisted laser desorption/ionization–time of flight mass spectrometry, liquid chromatography time-of-flight mass spectrometer, NMR and Raman analysis (Supplementary Figs. S6–11 and Supplementary Table S3). Based on the above results, we proposed the possible synthetic steps of $C_3N_5$ (Supplementary Fig. S12).

## Dipole moment-induced spontaneous polarization in $C_3N_5$

By introducing nitrogen-rich triazole groups into the carbon nitride framework, asymmetry of the structure occurs, which will bring a spontaneous polarization effect to regulate the migration and utilization of charge carriers. This enhanced polarization field of $C_3N_5$ was confirmed by piezoresponse force microscopy (PFM). As shown in Supplementary Fig. S13, an obvious resonance peak near 215 kHz reflects the piezoelectric vibration applied by different voltages for $C_3N_5$, and the amplitude shows linear piezoelectricity with voltage. The phase-voltage hysteresis loop of $C_3N_5$ (blue line of Fig. 2e) with a 180° change under a 2.5 V direct current bias field displays a distinct hysteresis with local polarization switching behavior, while the amplitude-voltage butterfly loop (red line of $C_3N_5$) has an amplitude of approximately 2.4 nm, which is much larger than the value of 1.1 nm for $C_3N_4$ (red line of $C_3N_4$). In amplitude-voltage loops, $d_{33}$ (effective piezoelectric coefficient) can be calculated as[30]:

$$d_{33} = \frac{D - D_I}{V - V_I}$$

where $D_I$ and $V_I$ are the displacement and voltage, respectively, at the intersection of the loop. $D$ and $V$ are the respective values at different points of the loop. Using the above formula, the maximum effective $d_{33}$ coefficients of $C_3N_4$ and $C_3N_5$ were approximately calculated as 0.196 and 1.548 nm/V, respectively (Supplementary Fig. S14). To further identify surface charge modulation, the surface piezoelectric potential distribution with the surface morphologies of $C_3N_5$ and $C_3N_4$ was evaluated by Kelvin probe force microscopy (KPFM) (Fig. 2f and Supplementary Fig. S15). Upon illumination, the KPFM images become brighter for n-type carbon nitride[31]. The results agree well with the contact potential difference (CPD) increases for n-type semiconductors under light. The range of CPD for $C_3N_5$ under dark conditions is approximately 859–888 mV, which is apparently higher than the range of 607–635 mV for $C_3N_4$; that for $C_3N_5$ under light conditions is approximately 908–946 mV, which is also higher than the range of 620–649 mV for $C_3N_4$. Moreover, $C_3N_5$ with light irradiation exhibits an increase in the average surface potential of ≈44.42 mV relative to that without light, while $C_3N_4$ with light shows an increase of ≈10.41 mV (Fig. 2f). This is because the spontaneous polarization induced by the dipole field of $C_3N_5$ amplifies the directional charge transfer upon light irradiation. As previously reported, compared to nonpolar materials, the surfaces of polar materials with dipole moments exhibit more significant upward band bending[31], which effectively inhibits charge recombination upon irradiation with modulated light, and the overall surface potential of the whole material is increased (Supplementary Fig. S15). Furthermore, the huge variation of ΔCPD upon distance indicates the uneven spatial distribution of surface band bending, which may be correlated with the local polarization structures of $C_3N_5$ and $C_3N_4$. The piezoelectric potential distribution of the samples was simulated by COMSOL Multiphysics software and the finite element method (Supplementary Fig. S16). Other PFM characterization results of amplitude error, amplitude and phase images are shown in Supplementary Fig. S17. These results fully confirm that the introduction of a dipole field can effectively enhance the local inherent piezoelectric properties of $C_3N_5$. Under light, the surface potential and

local polarization of triazole-based $C_3N_5$ improve the directional migration of photogenerated carriers.

## Dipole field effect in photocatalytic $H_2O_2$ production

To investigate the role of dipole field effect in photocatalytic $H_2O_2$ production, $C_3N_4$ and $C_3N_5$ with different dipole moments were placed in a reactor under both ultrasonic (Us) force and visible (Vis) light conditions (Supplementary Figs. S18 and S19). As shown in Fig. 3a, b, the performance of $C_3N_5$ and $C_3N_4$ in various scenarios at 60 min follows the sequence $C_3N_5$/Us/Vis > $C_3N_4$/Us/Vis > $C_3N_5$/Vis > $C_3N_4$/Vis > $C_3N_5$/Us > $C_3N_4$/Us. Note that there is only a slight increase in the $H_2O_2$ yield for $C_3N_5$ relative to $C_3N_4$ under Us, while there is a certain increase in the $C_3N_5$ yield under Vis light compared to under Us only. Interestingly, when Us and Vis light are simultaneously applied, the $H_2O_2$ yield of $C_3N_5$ (1.24 mmol $g^{-1}$ $h^{-1}$) is 2.75 times higher than that with light only, 5.25 times higher than that with Us only, and 2.54 times higher than the $C_3N_4$ yield with Us and Vis light. Moreover, the processes of $H_2O_2$ production over $C_3N_5$ for a wide pH range (pH 1–11) were also explored (Supplementary Fig. S20), and the optimum pH is 3. For comparison, the performance of $C_3N_4$ and $C_3N_5$ without EtOH (pH = 3) and in pure water (pH = 7) was also evaluated, and similar trends were observed (Supplementary Figs. S21 and S22). Negligible $H_2O_2$ is observed from sonication and stirring-only experiments (Supplementary Figs. S23 and S24). The apparent quantum efficiency (AQE) of $C_3N_5$ was calculated at specific wavelengths and shown to approximately match the UV-Vis spectrum (Supplementary Fig. S25). These results indicate that the dipole field in $C_3N_5$ can effectively promote the $H_2O_2$ production efficiency.

Note that the yield of $H_2O_2$ is determined by both the formation rate ($k_f$) and decomposition rate ($k_d$) (Fig. 3c). The reaction kinetic equation can be expressed as [$H_2O_2$] = ($k_f/k_d$) {1-exp(-$k_d$ t)}, where $k_f$ and $k_d$ were obtained by assuming corresponding zero-order and first-order kinetics (Supplementary Figs. S26 and S27)[27]. These analyses show that $C_3N_5$/Us/Vis has the highest $k_f$ and lowest $k_d$, suggesting a higher overall $H_2O_2$ yield than in the other conditions. To identify the role of $O_2$ in $H_2O_2$ production, different $O_2/N_2$ atmospheres were investigated. As shown in Fig. 3d, the $H_2O_2$ production efficiency is higher in the presence of $O_2$ during the reaction, suggesting an ORR mode under the $C_3N_5$/Us/Vis system. The outstanding $2e^-$ ORR of $C_3N_5$ for $H_2O_2$ production with high selectivity (92%) was further evaluated by rotating ring-disk electrode (RRDE) analysis, as shown in Supplementary Fig. S28[4,32].

The expansion of practical applications requires many factors, including high activity, an appropriate medium, and good stability. The $H_2O_2$ yield under different solutes and wavelength spectra are shown in Fig. 3e and Supplementary Fig. S29. Note that the yield in water (pH = 3) under simulated sunlight irradiation can reach approximately 3.8 mmol $g^{-1}$ $h^{-1}$ $H_2O_2$. Special attention needs to be paid to the fact that the current yield shows a high value among most of the reported carbon nitride-based photocatalysts and piezocatalysts and is even higher than that of most piezo-photocatalysts (Supplementary Fig. S30 and Supplementary Table S4). The data show the superior performance of triazole-based $C_3N_5$ with a dipole field in $H_2O_2$ photosynthesis. In addition, the as-prepared $C_3N_5$ yield remains stable for four cycles (Supplementary Fig. S31), and the XRD pattern and XPS spectra of used $C_3N_5$ show almost no change compared with fresh $C_3N_5$ (Supplementary Figs. S32 and S33). Finally, the produced $H_2O_2$ under pure water conditions was applied for on-site inactivation of *E. coli*. As shown in Fig. 3f and Supplementary Fig. S34, with increasing reaction time, the viability of *E. coli* decreases, and the quantity of viable bacteria markedly plummets when using the solution at a reaction time of 60 min.

## Photogenerated charge migration behavior with dipole moment

Compared with $C_3N_4$, the higher performance of $C_3N_5$ for $H_2O_2$ production under Us/Vis conditions means that rapid electron migration

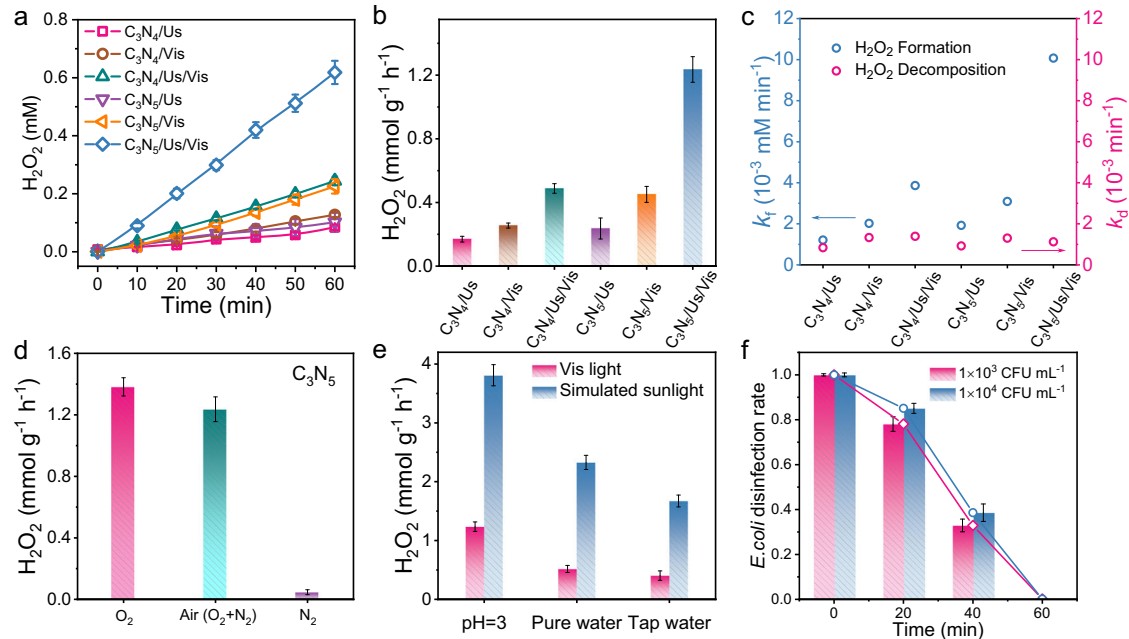

**Fig. 3 | Evaluation of H₂O₂ photosynthesis. a** Time profiles of photocatalytic H₂O₂ production by C₃N₅ and C₃N₄ in various scenarios. Experimental conditions: catalyst (0.5 g L⁻¹) with 10 vol% EtOH under Us and Vis light (λ ≥ 420 nm), T = 25 °C, water (pH = 3). **b** Corresponding histograms of the H₂O₂ yield at 60 min. **c** Comparison of the H₂O₂ formation rate constant ($k_f$, blue circle) with the H₂O₂ decomposition rate constant ($k_d$, red circle). **d** Effect of dissolved oxygen on H₂O₂ production for C₃N₅/Us/Vis in 1 h. **e** Photocatalytic H₂O₂ production over C₃N₅ with Us in pH = 3 water, pure water and tap water containing 10 vol% EtOH under Vis light (λ ≥ 420 nm) and simulated sunlight. **f** Real-time *E. coli* disinfection using generated H₂O₂. Error bars indicate the standard deviation from three measurements.

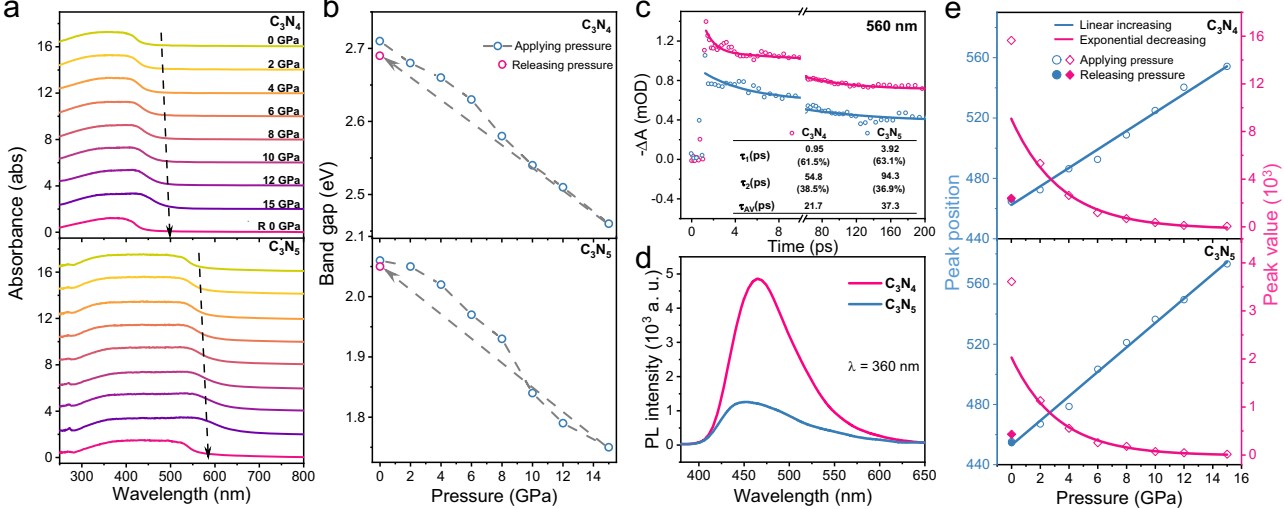

**Fig. 4 | Charge separation dynamics of C₃N₄ and C₃N₅. a** In situ pressure-dependent UV–Vis absorption spectra. **b** Bandgap variation with pressure. **c** Time profiles of TA at 560 nm. **d** Solid-state PL spectra at an excitation wavelength of 360 nm. **e** PL peak position (blue line) and PL peak intensity (red line) as a function of pressure at an excitation wavelength of 355 nm.

occurs during the photocatalytic process. To investigate the charge migration behavior, the redox band positions of C₃N₄ and C₃N₅ were investigated. The basic energy band structures of C₃N₄ and C₃N₅ were analyzed by Tauc plots and Mott-Schottky plots (Supplementary Figs. S35–37). Usually, the band positions of piezoelectric materials will tilt under an external force, which strengthens charge migration[5,33]. The in situ pressure-dependent UV–vis absorption spectra were investigated to trace the force-induced absorption edges of C₃N₄ and C₃N₅ (Supplementary Fig. S38). As shown in Fig. 4a, with increasing pressure from 0 to 15 GPa, the absorption edges of the two samples show obvious redshifts. The bandgap of C₃N₄ decreases from 2.71 to 2.46 eV, while that of C₃N₅ decreases from 2.06 to 1.75 eV

(Supplementary Fig. S39). When the pressure is released to 0 GPa, the bandgaps of C₃N₄ and C₃N₅ decrease to 2.69 and 2.05 eV, respectively (Fig. 4b), suggesting that the intrinsic bandgap of carbon nitride is changed by applying pressure. Density functional theory was further used to simulate the change in the bandgap for the two samples with pressure (Supplementary Fig. S40). The bandgaps of C₃N₄ without and with pressure are 2.58 and 1.646 eV, respectively, while those of C₃N₅ are 1.91 and 1.46 eV, respectively. This trend is similar to the results of in situ pressure-dependent UV–vis absorption spectra. These results indicate that applying pressure can effectively modulate the local energy band, which facilitates the migration and transport of photocatalytic charge carriers.

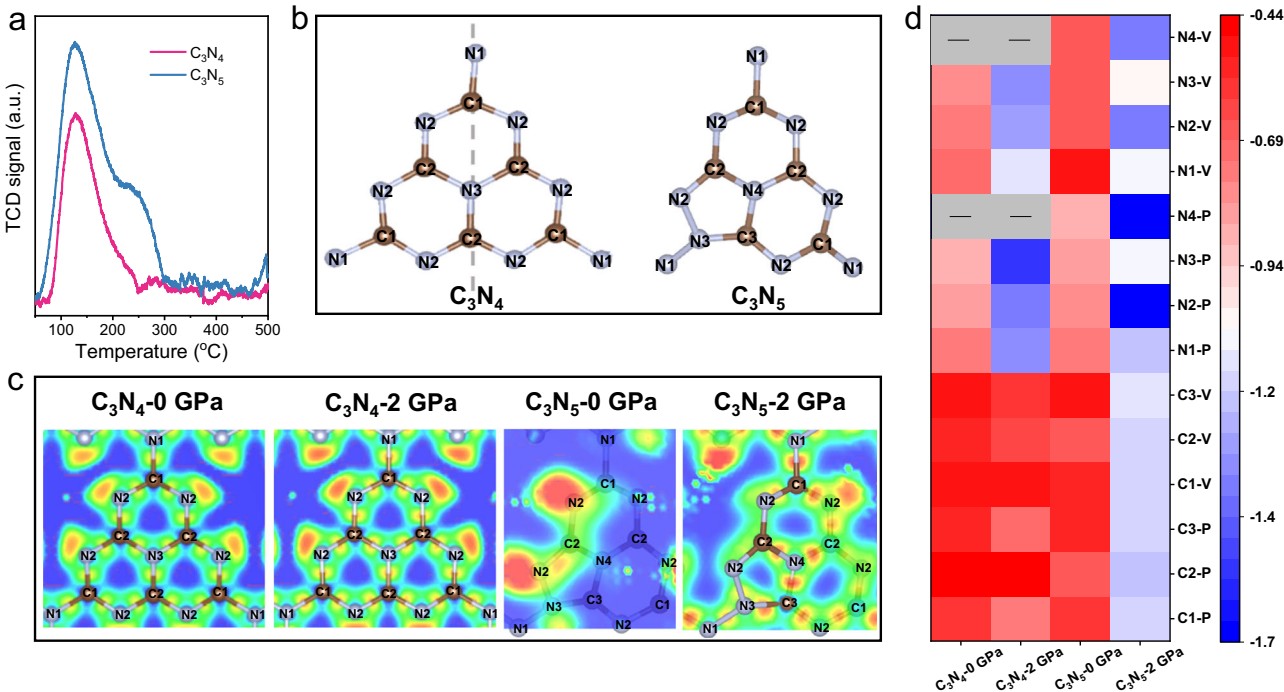

**Fig. 5 | Reactive sites in photocatalytic H₂O₂ production. a** O₂-TPD curves. **b** Molecular structure models with atomic numbering. **c** ELF with 0 and 2 GPa (isolevel = 0.8). **d** Hot spot diagram of vertical (V) and plane (P) adsorption energies of O₂ for each C or N atom at 0 and 2 GPa.

To understand the charge carrier behavior in $C_3N_4$ and $C_3N_5$ with different dipole moments, the dynamics of the charge carriers were investigated by ultrafast transient absorption (TA) and photoluminescence (PL) spectroscopy. Pseudocolor TA plots and TA curves of $C_3N_4$ and $C_3N_5$ are shown in Supplementary Figs. S41 and S42. Both samples exhibit typical TA bands in the Vis light region of 450–750 nm upon 405 nm laser excitation. Compared with $C_3N_4$, $C_3N_5$ presents slightly decreased absorption. The time profiles of TA at 560 nm for $C_3N_4$ and $C_3N_5$ were fitted by two exponential functions (Fig. 4c), and fitting results with a short electron lifetime ($\tau_1$) = 0.95 ps (61.5%) and a long electron lifetime ($\tau_2$) = 54.8 ps (38.5%) for $C_3N_4$ and with $\tau_1 = 3.92$ ps (63.1%) and $\tau_2 = 94.3$ ps (36.9%) for $C_3N_5$ were observed. The short lifetime and long lifetime usually correspond to electrons trapped at shallow and deep sites before charge recombination[34]. Notably, $C_3N_5$ has longer lifetimes than $C_3N_4$, suggesting that the introduced triazole group in $C_3N_5$ acts as a trap site of electrons[35], and these photoexcited electrons are rapidly transferred via the dipole field to active sites in the triazole group. The solid-state PL spectra reveal that $C_3N_5$ has a lower PL intensity than $C_3N_4$ (Fig. 4d). The higher PL quenching efficiency of 73.8% in $C_3N_5$ indicates inhibition of charge recombination by the formed dipole field. To give a visual view, single-particle PL spectra were also investigated, and similar trends were obtained; detailed discussions are described in the Supplementary Information (Supplementary Figs. S43–46). The charge carrier migration properties were further demonstrated by electrochemical behaviors, including piezo-photocurrents and electrochemical impedance spectroscopy (EIS) (Supplementary Figs. S47).

To monitor the charge migration process in photoexcited $C_3N_5$ under an external force, in situ pressure-dependent PL spectroscopy was performed. With gradually increasing pressure from ambient to 15 GPa, both $C_3N_4$ and $C_3N_5$ display similar changes from initial blue to green and finally to colorless at high pressure (Supplementary Figs. S48 and 49). These results are consistent with the chromaticity diagram of the Commission Internationale de l'Eclairage (CIE) (Supplementary Fig. S50). As shown in Supplementary Fig. S51 and Fig. 4e, as the pressure increases, the PL peak positions of the two samples

show an obvious redshift and the peak intensity of $C_3N_5$ is significantly lower than that of $C_3N_4$. When the pressure is released to 0 GPa, the peak position recovers, whereas the peak intensity decreases because the destruction of the structure under applied pressure weakens the luminescence of the material[36]. These results confirm that the inhibition efficiencies for charge recombination in photoexcited $C_3N_4$ and $C_3N_5$ increase with increasing external pressure, and the charge separation efficiency for $C_3N_5$ is higher than that for $C_3N_4$. Therefore, the dipole field in $C_3N_5$ improves the charge migration behavior.

### Reactive sites in $C_3N_5$ for H₂O₂ production

In the ORR, to produce H₂O₂, O₂ adsorption on the surface of the catalyst is the first step. The O₂ temperature-programmed desorption (O₂-TPD) curve reveals that $C_3N_5$ has a higher O₂ adsorption (1.07 mmol g⁻¹) than $C_3N_4$ (0.54 mmol g⁻¹) (Fig. 5a). The molecular structure models with atomic numbering are shown in Fig. 5b, containing a symmetric structure of $C_3N_4$ and an asymmetric structure of $C_3N_5$. The electron localization function (ELF) was computed to elucidate the profiles of their localized electron distribution. As shown in Fig. 5c, the electron clouds of the two samples without pressure are smaller than those with pressure. In particular, the active charge distribution of $C_3N_4$ has almost no change, while that of $C_3N_5$ is changed from the N2 site on the left side to the N2 and N4 sites on the right side from 0 to 2 GPa. This phenomenon indicates that the introduction of the triazole ring induces asymmetry to boost the change in polarization sites with pressure.

Supplementary Fig. S52 shows the electron distribution between O₂ and $C_3N_4/C_3N_5$ in vertical and plane adsorption. A larger electron cloud and a stronger electron distribution occur between O₂ and $C_3N_4/C_3N_5$ with pressure, and stronger covalent bonds are generated, as manifested by the high charge density at this interface. For the ORR process, a longer bond length of O₂ indicates that it is easier to activate, while a shorter bond length between O₂ and the catalyst atom means that O₂ is more likely to be absorbed on the catalyst. As shown in Supplementary Fig. S52, the O-O and C/N-O bond lengths for all samples mostly display an increasing and decreasing trend with

pressure compared to those without pressure. The strong charge interaction between adsorbed $O_2$ and N in triazole on $C_3N_5$ was explored by the charge density difference in Supplementary Fig. S53. The plane-adsorbed $O_2$ on the C or N atoms of $C_3N_5$ has the strongest adsorption energy ($E_{ads}$) and highest charge transfer number ($|e|$) with pressure. We also simulated the adsorption energy of each C and N atom of the two samples in different scenarios (Fig. 5d and Supplementary Table S5). The N4 sites in triazole on $C_3N_5$ exhibit the strongest adsorption energy for $O_2$ in the plane adsorption with pressure. These results indicate that asymmetric triazole-based $C_3N_5$ can drive spontaneous polarization charges under a certain pressure, and the N4 sites in the triazole of $C_3N_5$ are the key ORR reactive sites.

The adsorption configuration of $O_2$ on the surface of carbon nitride is also crucial for the ORR. It is generally classified into three types: Pauling-type (end-on), Griffiths-type (side-on) and Yeager-type (side-on)[3,37]. The end-on $O_2$ adsorption configuration can minimize O-O bond breaking, leading to suppression of the $4e^-$ ORR and a highly selective $2e^-$ ORR. Herein, different $O_2$ adsorption configurations of the two samples were simulated by using DFT calculations. Supplementary Fig. S54 shows that $C_3N_4$−0 GPa displays side-on $O_2$ adsorption, while $C_3N_4$−2 GPa, $C_3N_5$−0 GPa and $C_3N_5$−2 GPa show end-on $O_2$ adsorption, indicating a highly selective $2e^-$ ORR for $C_3N_5$.

### Photocatalytic $H_2O_2$ production pathway via $C_3N_5$

Based on the above premise, the reaction dynamics and thermodynamics of ORR pathways on $C_3N_5$ with pressure were investigated via the Gibbs free energy ($\Delta G$) and the configurations of intermediates (Fig. 6a and Supplementary Fig. S55). The results indicate that $C_3N_5$ with pressure has the lowest free energy for $O_2$, which is accompanied by two reactive oxygen intermediates in sequence (*$O_2^-$ to *OOH) and then further proton coupling to produce $H_2O_2$. To further explore the reaction mechanisms and active species, different sacrificial agents were separately added to the initial solution (Supplementary Fig. S56), demonstrating that superoxide radicals (*$O_2^-$) mainly contribute to $H_2O_2$ production. The production of *$O_2^-$ was also proven by EPR spectroscopy, as shown in Supplementary Fig. S57.

To monitor the intermediates during the photocatalytic $H_2O_2$ production process via $C_3N_5$ with Us, in situ EPR experiments (Fig. 6b and Supplementary Fig. S58) were used to reveal their reaction dynamic processes. DMPO was used to capture the in situ generated *$O_2^-$ and *OOH. As shown in Fig. 6c, d, after Us and Vis light are applied, the *$O_2^-$ signal is observed at 30 s and continues to grow to approximately 118 s, with an optimum concentration of $9.27 \times 10^{-6}$ M. Immediately afterward, the *OOH signal appears, with essentially the same concentration as *$O_2^-$. Then, the concentration of *OOH sharply decreases to $3.76 \times 10^{-6}$ M from 118 to 200 s, which is due to *OOH with a short lifetime not being stable enough and being rapidly converted to $H_2O_2$ by protonation, suggesting that this process is an indirect $2e^-$ ORR in $H_2O_2$ production[14,38]. The detailed full spectra with a sweep time of 2.5 s per sample are provided in Supplementary Fig. S59, and simulated EPR spectra are also provided to confirm the above dynamic production processes of the free radicals (Supplementary Fig. S60).

On the basis of the above discussions, a mechanism of dipole field-induced spontaneous polarization to promote photocatalytic $H_2O_2$ production is proposed (Fig. 6e). Primarily, due to the asymmetric unit of triazole in $C_3N_5$, a spontaneous dipole field is formed in the $C_3N_5$ plane. Under Vis/Us conditions, this dipole field forces the photogenerated electrons and holes to undergo directional migration. Then, with the assistance of Us, the solution oxygen is easily adsorbed on the surface of N4 atoms in the triazole unit with Pauling-type (end-on) binding. Finally, the adsorbed $O_2$ is step-by-step reduced through an indirect $2e^-$ transfer pathway with intermediates of *$O_2^-$ and *OOH to form $H_2O_2$ on the surface of the triazole N4 sites. The opposite holes are quenched by EtOH to provide sufficient electrons to balance the whole reaction.

## Discussion

In summary, we report the polymerization of a triazole and triazine framework to form nitrogen-rich $C_3N_5$, which generates a strong dipole field due to its asymmetric structure. Compared with traditional carbon nitride ($C_3N_4$), the as-prepared $C_3N_5$ displays a marvelous $H_2O_2$ yield (3809.5 μmol $g^{-1}$ $h^{-1}$) with high $2e^-$ transfer selectivity (92%) under simulated sunlight and ultrasonic force conditions, which exceeds most photosynthetic processes based on carbon nitride or piezophotocatalytic processes. The introduced triazole group contributes to the above superior artificial photosynthesis of $H_2O_2$ in $C_3N_5$. First, this triazole group brings an asymmetric structure to generate a dipole field with spontaneous polarization, which forces photoinduced charges to undergo directional separation to active sites. Second, the N atoms in the triazole group act as ideal active sites, and $O_2$ and electrons are both easily trapped at these N active sites via the dipole field under the external force, resulting in a rapid $H_2O_2$ production process with an indirect $2e^-$ transfer pathway. This kind of structural engineering generates a dipole field to harness the migration of photogenerated carriers and provides a feasible strategy to improve photosynthesis efficiency, and the present innovation is ideally suited as a fundamental approach for catalyst molecular design.

## Methods

The material, instruments, other experiments and characterizations are discussed in Supplementary Information (Supplementary Texts S1–S12).

### Preparation of nitrogen-enriched carbon nitride

Brown $C_3N_5$ was prepared by thermal polymerization[18]. About 2.0 g of 3-amino-1,2,4-triazole (3-AT) powders was put into an $Al_2O_3$ crucible, and the crucible was covered with an $Al_2O_3$ cover to keep a half-cover state. The crucible was then heated to 500 °C in a muffle oven at a rate of 5 °C/min, kept at 500 °C for 3 h, and then cooled down to room temperature. $C_3N_4$ was obtained by heating 1.0 g melamine at 500 °C for 2 h in a semi-closed system to prevent sublimation of melamine[39]. Prior to photosynthesis, two samples were subjected to ultrasound treatment for 30 min to form nanosheets. Carbon nitride structures at different temperatures, including 200, 300 and 400 °C using 3-AT (viz. CN-200 °C, CN-300 °C and CN-400 °C) were also synthesized for comparison.

### Photocatalytic $H_2O_2$ production under ultrasonic force

In this, 20 mg of catalyst was added to 40 mL of pure water containing ethanol (10 vol%) at pH = 3. The catalyst was dispersed by ultrasonication for 10 min, and air was bubbled through the solution for 10 min. The reactor was kept at 25 ± 0.5 °C with cooling circulating water and was irradiated at $\lambda \geq 420$ nm using a 300 W Xe lamp (PLS-SXE300D, Beijing Perfectlight Technology Co., Ltd) with a light intensity of 100 mW $cm^{-2}$, and simultaneously subjected to ultrasonication by an ultrasonic cleaner (40 kHz, 100 W, Jielimei, Kunshan, China). The light-only experiments were placed under a xenon lamp with stirring. The concentration of $H_2O_2$ was measured by the KI colorimetric method[40]. One milliliter of freshly prepared KI reagent A (0.4 M KI, 0.05 M NaOH, $1.6 \times 10^{-4}$ M $(NH_4)_6Mo_7O_{24} \cdot 4H_2O$) and 1 mL of reagent B (0.1 M $KHC_8H_4O_4$) were mixed with 1 mL of the above samples. The absorbance of the above mixture was measured at 350 nm by a UV–Vis spectrophotometer (JASCO V-770, Japan). To study the effects of gases, different gases, including $N_2$, $O_2$ and air, were bubbled through the solution for 15 min to conduct subsequent experiments.

### In situ high-pressure PL and UV–Vis absorption spectra

High-pressure experiments were performed using a symmetric diamond anvil cell (DAC). A pair of ultra-low fluorescence diamonds with an anvil surface of 400 μm diameter was used to generate pressure for the in situ high-pressure PL experiments. A T301 stainless steel gasket

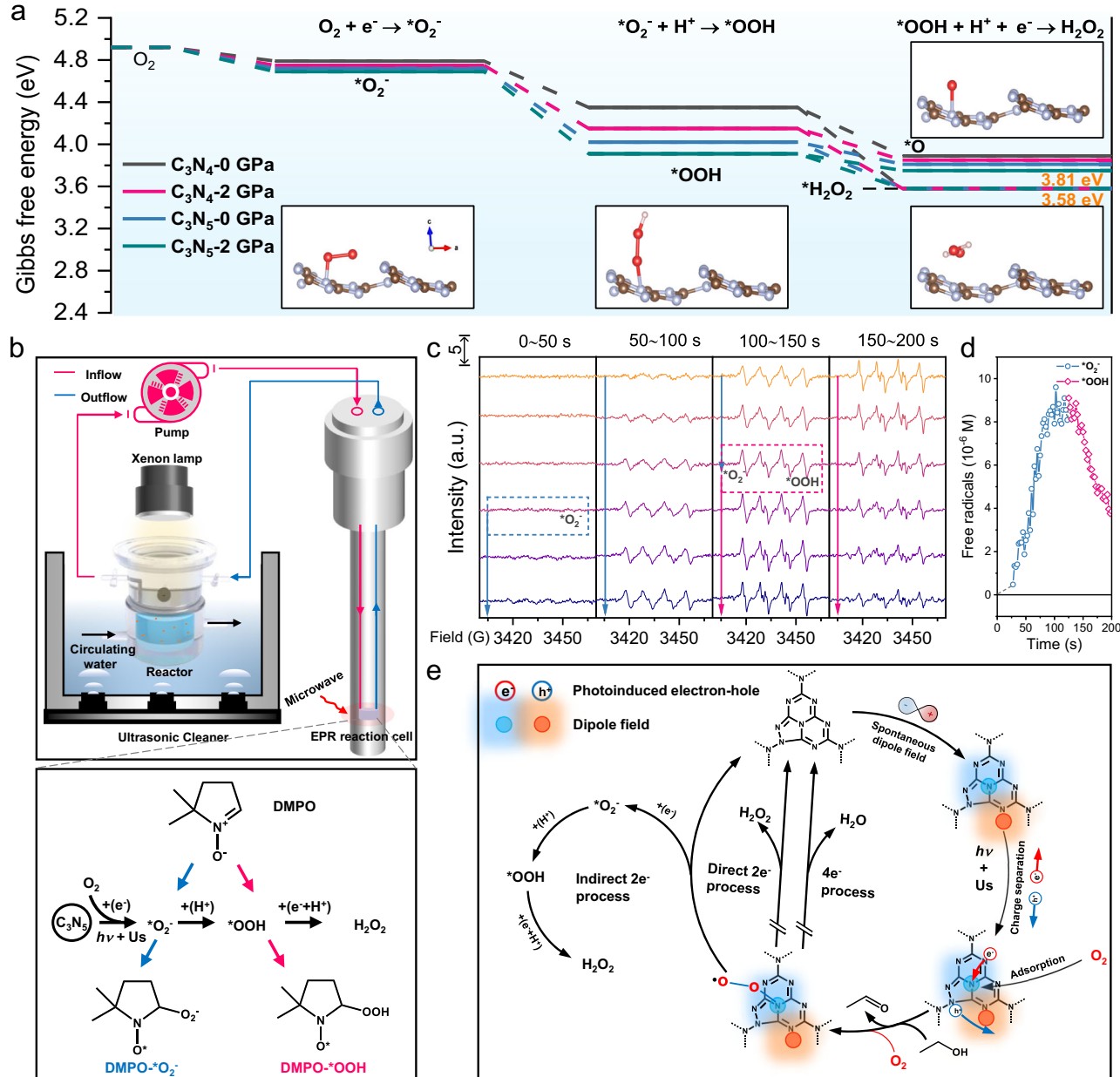

**Fig. 6 | Reaction dynamics, thermodynamics and mechanism of photocatalytic H₂O₂ production. a** ΔG diagram of C₃N₄ and C₃N₅ in the ORR reaction process with and without pressure. The insets show the transition state in the ORR of C₃N₅ with pressure, and the brown and gray colors represent carbon and nitrogen. **b** In situ EPR system and principle of 5,5-dimethyl-1-pyrroline (DMPO) capturing *O₂⁻ and *OOH. **c, d** In situ EPR spectra (**c**) and concentration and lifetime (**d**) of different free radicals during H₂O₂ production via C₃N₅/Us/Vis. **e** Proposed reaction mechanism.

pre-indented to the thickness of 45 μm was laser drilled to obtain a 150 μm diameter hole for loading the sample and a ruby ball. The ruby fluorescence technique was adopted for the pressure calibration in connection with the shift of the R1 line of the ruby fluorescence. The silicon oil was applied as a pressure-transmitting medium around the sample. All the high-pressure experiments were conducted at room temperature.

The high-pressure PL measurements were performed by using a combined homemade optical measurement system (Supplementary Fig. S61). The pressure-dependent PL spectra were measured by a semiconductor laser with an excitation wavelength of 355 nm. Note that the effects of different excitation laser intensities and luminous fluxes on the obtained PL intensity of carbon nitride were avoided by fixing all the parameters during each high-pressure PL experiment. Absorption spectra were studied in the exciton absorption band

region using a deuterium-halogen light source. High-pressure PL and absorption spectra of carbon nitride were recorded with an optical fiber spectrometer (Ocean Optics, QE65000).

## Data availability
The data that support the plots within this paper and other findings of this study are available from the corresponding author upon reasonable request.

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

## Acknowledgements

M.Z. acknowledges the support of the National Science Foundation of China (No. 22322604), the Guangdong Basic Applied Basic Research Foundation (No. 2020B1515020038), the Science and Technology Program of Guangzhou (No. 202201020545) and Pearl River Talent Recruitment Program of Guangdong Province (No. 2019QN01L148). Y.Y. acknowledges the support of the National Science Foundation of China

(No. 22276116). S.C. acknowledges the support of the Guangdong Basic and Applied Basic Research Foundation (Nos. 2021A1515110907 and 2023A1515011935). The authors also acknowledge the support for NMR analysis by Dr. A. Guan from the Institute of Chemistry, Chinese Academy of Sciences (CAS), and Dr. Y. Ye from JEOL Beijing.

## Author contributions

M.Z. and Z.L. conceived the experiments, analyzed the data and wrote the paper with the inputs from all authors. M.Z. supervised the project. Z.L. fabricated the materials and carried out the main experiments. Y.Y.Z. analyzed experimental data. K.W. and B.Z. carried out the high-pressure experiments. Y.T.Z. made the theoretical analyses. S.C. did the electrochemical experiments. Y.Y. did the bio-experiments and theory data. W.J. did and analyzed the in situ data and writing—review & editing. L.C. did mass spectrometry analysis. All authors joined in to review and edit the paper.

## Competing interests

The authors declare no competing interests.
