## [Peer Review File · Nature Communications]

REVIEWER COMMENTS

Reviewer #1 (Remarks to the Author):

Li et al developed in this work a piezo-photocatalyst based on carbon nitride for photocatalytic oxygen reduction to hydrogen peroxide. The authors aim to fabricate a C₃N₅ network containing dipoles that boosts the charge separation efficiency and employ a very wide range of techniques to prove their hypothesis. The prepared material works indeed better than the reference and displays certain piezoelectricity, however the hypothesis of the authors relies on having a C₃N₅ network which is not supported by the experimental evidence.

The authors don't mention in the manuscript any aspect regarding the synthesis and its rationale, just briefly in the SI, a triazole-based molecule is employed as starting monomer. Such molecules have been employed previously for the synthesis of carbon nitride observing a higher release of ammonia and nitrogen gas (by means of GC-MS) rather than the formation of a C₃N₅ (J. Mater. Chem. A 2017, 5, 8394). The differences in characterization between the so called C₃N₄ and C₃N₅ are less than minimal, for instance it is pretty much impossible to distinguish between both N1s spectra other than the y-scale is not normalized, while in previous C₃N₅ reports the difference was very substantial (J. Am. Chem. Soc. 2019, 141, 5415-5436). In the case of the solid state NMR, 3 out of the 5 chemical shifts for ¹⁵N cant be distinguished from the background signal.

The prepared material does present defects and a substantially narrower band gap which could be contributing to the enhanced photocatalytic performance. Claiming that such performance is higher than the state of the art, however, is quite misleading as the performance strongly depends also on reactor design so one could just make such claim if other catalysts have been tested under sonication and illumination with the same intensity (and the reactor design shown here is quite unconventional).

Additionally, the 2e⁻ selectivity has been assessed electrochemically by means of an RRDE which substantially different to a photocatalytic scenario, one depending on surface area and adsorption desorption and the other one depending more heavily in energy band position. Average quantum yield for instance would be more useful.

Following this reasons i don't think this manuscript can be considered for publication in Nature Communications and i recommend the authors to either reword the discussion acknowledging the ambiguity of the structure, or try to elucidate further the structure by means of mass spectrometry or elemental analysis, which is very useful to prove the structure of melem analogues (RSC Adv. 2021, 11, 38862)

Reviewer #2 (Remarks to the Author):

The authors have demonstrated the importance of the internal dipole field in nitrogen-rich carbon nitride photocatalysts and its impact on charge separation and H₂O₂ production. This work is of significant contribution to the field and deserves to be published in this journal after revisions addressing the following comments and questions:

1. Clarify if the polarization induced by the dipole field is triggered by an external force, such as pressure or ultrasonication, or if it is a physical property of nitrogen-rich carbon nitride.
2. In Figure 3b, C₃N₄ produces H₂O₂ only with sonication despite lacking an internal dipole field. Recent reports suggest that C₃N₄ nanosheets exhibit in-plane piezoelectricity. Were the samples sonicated to form nanosheets before conducting photosynthesis?
3. Clarify if the H₂O₂ production in C₃N₅ under N₂ gas is due to oxygen pre-adsorbed on C₃N₅, and why the H₂O₂ production rate slightly decreases under air. Sonication of water can produce H₂O₂ through the sonolysis of H₂O molecule. Please carry out more control experiments to confirm how much of H₂O₂ is generated from the sonolysis of water alone.
4. In Figure 4e, the peak strength is reduced after releasing pressure compared to before applying pressure. Clarify how the pressure affects the material's structure.
5. Conduct scavenger tests with different concentrations to obtain more reliable quenching results.
6. Provide more details on the light-only experiment, such as whether it was conducted under stirring, and add a separate stirring experiment without light as a control. Also, clarify the differences between ultrasound and stirring.
7. Provide more details on how the authors conducted in-situ pressure-dependent PL spectra, such as how they achieved the high target pressure on quartz.
8. Evaluate the electron transport capacity of C₃N₅ and C₃N₄ using photocurrent and impedance measurements.
9. Include more references related to polarization-induced exciton dissociation in polymeric photocatalysts, such as Chem Catal 2, 1734-1747; Chem Catalysis 2 (7), 1517-1519.

Reviewer #3 (Remarks to the Author):

Dipole field in nitrogen-enriched carbon nitride Photosynthesis of H₂O₂ from O₂ and water is an ideal way to convert and store solar energy as well as the production of useful chemicals, because it can make full use of the photogenerated electrons and holes. Li et al. reported the photocatalytic H₂O₂ production on nitrogen-rich C₃N₅. This present work has very fruitful characterizations and mechanism investigations. The structural engineering concept is novel and inspiring, and the results are rich and convincing. I think it can be published in Nature Communications after following issues being addressed.

1. There are conceptual problems in Figure 1c. There always exists a surface band bending for a semiconductor, whose direction depends on the semiconductor type (n or p) and the exposed environment [Chem. Rev. 2012, 112, 5520-5551]. So the flat potential throughout the bulk and surface in (i) is not scientifically accurate. It is suggested to present a n-type case band bending, considering the n-type nature of the carbon nitride in this work. The concept demonstrated in (ii) is also problematic. Besides the band bending issue mentioned above, the transfer direction of charge carriers is not correct. As the energy level shown here generally represents the energy of electrons, the electrons should transfer from the high position to the low position of the energy level lines, and holes migrate to the opposite direction. In terms of the effect of dipole moment on the surface band bending, please refer to the paper [Angew. Chem. 2020, 132, 945-952].
2. In Figure 2f, "an increase in the average surface potential" is used to characterize the effect of light on the surface potential. Yet it is not rational because as we can see, the difference between the lowest potential and the highest potential with light on C₃N₅ is about 940-890=50 mV, even larger than the average increase of 30.61 mV. Actually, the CPD difference before and after light irradiation (Δ CPD) reflects the surface band bending extent [Chem. Soc. Rev. 2018, 47, 8238-8262]. It is suggested to present Δ CPD upon distance, and compare the two Δ CPD curves of C₃N₅ and C₃N₄. Moreover, the huge variation of Δ CPD upon distance indicates the uneven spatial distribution of surface band bending, which may give more insights into photogenerated carriers' kinetics if it can be correlated with the local structures of C₃N₅ and C₃N₄.
3. Solid state NMR is the most efficient and direct proof to investigate the detailed structure of carbon nitride-based materials. The author also used this technology to confirm the molecular structure. However, the present resolution of ¹⁵N NMR spectra is not convincing enough to get the conclusion. More precise characterizations should be performed to clearly demonstrate the structure.
4. In presence of alcohols as electron donors, some other carbon nitride-based catalysts show higher efficiency in H₂O₂ production, such as ACS Catal. 2020, 10, 24, 14380–14389, Nat. Commun., 2021, 12, 3701, Angew. Chem. Int. Ed., 2021, 60(48): 25546-25550, Chem Catal., 2022, 2(7): 1720-1733 etc. The Figure S20 should also include these activities in the latest publications.
5. To evaluate the efficiency of the catalyst, activity under monochromatic light should be performed to calculate the apparent quantum yield.

6. The in-situ pressure-dependent UV-vis absorption spectra characterization is very impressive. It may be inspiring to future works on the pressure-induced band gap modulation. It is suggested to provide a detailed picture of the set-up for this measurement to make it clearer to the readers.

7. Minor problems: The pictures for adsorption configuration in Figure 6a is fuzzy. Clear and distinct pictures should be provided. "The surface plane O₂ is adsorbed in a Pauling-type manner at the N active sites in C₃N₅" should be "On the surface plane, O₂ is adsorbed ..."

Manuscript ID: NCOMMS-23-06785

Title: Dipole field in nitrogen-enriched carbon nitride with external forces to boost the artificial photosynthesis of hydrogen peroxide

Reviewer #1 (Remarks to the Author):

Summary Comments: Li et al developed in this work a piezo-photocatalyst based on carbon nitride for photocatalytic oxygen reduction to hydrogen peroxide. The authors aim to fabricate a C_3N_5 network containing dipoles that boosts the charge separation efficiency and employ a very wide range of techniques to prove their hypothesis. The prepared material works indeed better than the reference and displays certain piezoelectricity, however the hypothesis of the authors relies on having a C_3N_5 network which is not supported by the experimental evidence.

Author response: We appreciate the reviewer's critical comments, which helped us to improve the overall quality of the manuscript significantly. By following the reviewer's comments, we have revised the manuscript in a point-by-point manner. In particular, we tried to add more details and clarify ambiguous claims. Our responses and modifications are as follows.

1. The authors don't mention in the manuscript any aspect regarding the synthesis and its rationale, just briefly in the SI, a triazole-based molecule is employed as starting monomer. Such molecules have been employed previously for the synthesis of carbon nitride observing a higher release of ammonia and nitrogen gas (by means of GC-MS) rather than the formation of a C_3N_5 (J. Mater. Chem. A 2017, 5, 8394). The differences in characterization between the so called C_3N_4 and C_3N_5 are less than minimal, for instance it is pretty much impossible to distinguish between both N1s spectra other than the y-scale is not normalized, while in previous C_3N_5 reports the difference was very substantial (J. Am. Chem. Soc. 2019, 141, 5415-5436). In the case of the solid state NMR, 3 out of the 5 chemical shifts for ^{15}N cant be distinguished from the background signal.

Author response: We appreciate the reviewer's critical comments. I am sorry that the previous version did not mention about any synthesis and its principles. In the revised version, we further demonstrate our structure using organic elemental analysis (UNICUBE-Elementar), Nuclear magnetic resonance (NMR, Bruker ADVANCE III 400 and 600 for solid-state NMR spectra and JNM-ECZ400R for liquid-state NMR spectra), Matrix-assisted laser desorption/ionization–time of flight mass spectrometry (MALDI-TOF-MS, Bruker Autoflex Speed TOF/TOF), liquid chromatography time-of-flight mass spectrometer (LC-TOF-MS, AB Sciex Triple TOF 5600), near-edge X-ray absorption finestructure (NEXAFS) at the beamline BL14W1 station of the Beijing Synchrotron Radiation Facility, China and Raman spectrometer (LabRAM HR Evolution spectrometer).

Indeed, the triazole-based molecule (3-amino-1,2,4-triazole, 3-AT) has been used for the synthesis of carbon nitride in a previous report [J. Mater. Chem. A 2017, 5, 8394]. The synthesis method described in this report is that 3-AT is treated with hydrochloric

acid, ammonium persulfate and alkali, after which the resulting mixture is subjected to high temperature heat treatment with N₂. However, our synthesis method is quite different from this report. We used a one-step pyrolysis method using air for the high temperature synthesis of triazole-based C₃N₅ backbone. Our methods are shown below: About 2.0 g of 3-amino-1,2,4-triazole powders was put into an Al₂O₃ crucible, and the crucible was covered with an Al₂O₃ cover to keep a half-cover state. The crucible was then heated to 500°C in a muffle oven at a rate of 5°C/min, kept at 500°C for 3 h, and then cooled down to room temperature in the muffle oven.

(Chemical structure of C₃N₅ in our work and previous work)

Also, we are not at all the same structure as the C₃N₅ report [J. Am. Chem. Soc. 2019, 141, 5415-5436] you mentioned before, the specific structure is shown in above figure, therefore, their structural characterization is also completely different. And they are also synthesized in a completely different process, which can be shown in previous report. Therefore, it is not appropriate to compare our work with the above two works.

(Table S3: in the revised Supplementary Information)

Table S3. Elemental composition of 3-AT, CN-200°C, CN-300°C, CN-400°C and C₃N₅ (CN-500°C) as determined by organic elemental analysis (C, N, H).

Sample	Nitrogen (wt%)	Carbon (wt%)	Hydrogen (wt%)	Carbon/ Nitrogen (Atomic ratio)
3-AT	66.8	28.8	6.76	0.503
	66.8	28.7	6.29	0.501
	66.7	28.7	5.54	0.502
CN-200°C	66.8	28.9	5.05	0.505
	66.5	28.7	5.77	0.504
	66.1	28.8	5.33	0.508
CN-300°C	64.8	29.1	3.95	0.524
	64.8	29.2	4.86	0.526
	64.7	29.3	4.84	0.528
CN-400°C	63.9	31.9	4.4	0.582
	63.6	31.7	4.79	0.581
	63.7	31.8	4.08	0.582
C ₃ N ₅ (CN-500°C)	63.1	33.1	3.21	0.611
	63.1	33.2	3.34	0.613
	63.0	33.0	3.42	0.611

Here, carbon nitride structures at different temperatures including 200, 300 and 400°C using 3-AT (viz. CN-200°C, CN-300°C and CN-400°C) were also synthesized for comparison. First, we evaluate the elemental ratio of C to N for all samples using organic elemental analysis. The CHN results of C₃N₅ (CN-500°C) confirm that the average weight percentages of C and N are 33.1% and 63.1%, respectively (Table S3). The average atomic rate of C/N is thus determined to be 0.611, which is very close to the theoretical value (0.60), confirming the successful synthesis of C₃N₅. Specifically, for 3-AT and CN-200°C, the elemental proportion of CNH has barely changed, which may be due to the fact that the boiling point of 3-AT is 244.9°C, implying that the chemical composition has not changed. For CN-300°C, CN-400°C and C₃N₅ (CN-500°C), with the increase of synthesis temperature, the percentage weight of N and H decreases gradually, this may be due to the possible release of ammonia and hydrogen during the synthesis process. The total mass fraction of 3-AT exceeded 100%, due to the water absorption of the sample.

(Figure S7: in the revised Supplementary Information)

Figure S7: MALDI-TOF-MS spectra (a) of CN-400°C; Major species (b) constituting the precipitate formed from CN-400°C. The solid sample was adopted for testing.

MALDI-TOF-MS was used to determine the chemical structure of intermediates. **Figure S7a** presents the whole MALDI-TOF-MS spectra of the intermediate of CN prepared at 400°C. Since the amino groups on the carbon nitride are easily charged with positive protons, we tested the intermediates in positive ion mode. The peaks with m/z of 64, 71, 96, 106, 127, 192, 196 and 211 Da are observed in the MALDI-TOF MS spectra of CN-400°C. Therefore, some possible molecular structures of the intermediates in the CN polymerization are listed in **Figure S7b**. Based on this, the above m/z can correspond to the eight positively ionized products of the four molecules as above. This implies the possible existence of these four intermediates. Other excess peak may be due to the effect of incomplete polymerization of 3-AT and other impurities.

(**Figure S8**: in the revised Supplementary Information)

Figure S8: LC-TOF-MS spectra (a) of H₂O and CN-400°C; Major species (b) constituting the precipitate formed from CN-400°C. DMSO was used as a solvent for CN-400°C, and DMSO was used as a control sample.

LC-TOF-MS was used to determine the chemical structure of intermediates. **Figure S8a** presents the whole LC-TOF-MS spectra of the intermediate of CN prepared at 400°C. We also tested the intermediates in positive ion mode, because the amino groups on the carbon nitride are easily charged with positive protons. The peaks with m/z of 43, 64, 71, 96, 106, 127, 192, 196 and 211 Da are observed in the LC-TOF-MS spectra of CN-400°C. Therefore, the positively ionization mode of these possible intermediates is shown in are listed in **Figure S8b**. These results are also very consistent with the results of MALDI-TOF-MS. This also demonstrates the possible existence of these four intermediates.

(**Figure S5**: in the revised Supplementary Information)

Figure S5: C K-edge NEXAFS spectra of C_3N_4 and C_3N_5 .

(**Figure 2c**: in the revised manuscript)

Figure 2c: N K-edge NEXAFS spectra of C_3N_4 and C_3N_5 .

Near-edge X-ray absorption finestructure (NEXAFS) analysis was also used to explore the chemical bonds of triazole-based C_3N_5 . The C K-edge NEXAFS spectra (**Figure S5**) of C_3N_5 and C_3N_4 show characteristic excitations including $1s \rightarrow \pi^*_{\text{out of plane C=C}}$ at ~ 285.2 eV and $1s \rightarrow \pi^*_{\text{N-C=N}}$ at 288.0 eV [Nat. Commun. 2014, 5, 3783]. Compared with C_3N_4 , C_3N_5 displays a blue shift of $1s \rightarrow \pi^*_{\text{N-C=N}}$ excitation, which is attributed to increased N-C=N bond strength upon the formation of triazole moiety. The N K-edge NEXAFS spectra (**Figure 2c**) of two samples show similar response at 399.3 and 402.5 eV, attributing to the $1s \rightarrow \pi^*_{\text{N-C=N}}$ and $\pi^*_{\text{C-N}}$ resonance. Note that, relative to C_3N_4 , triazole-based C_3N_5 displays a red shift of $1s \rightarrow \pi^*_{\text{N-C=N}}$ excitation and a new peak at 401.1 eV, which are attributed to the formation of triazole moiety and $1s \rightarrow \pi^*_{\text{heterocyclic N-N}}$ of triazole, respectively [Nat. Commun. 2014, 5, 3783]. These experimental findings provide strong evidence of the coexistence of triazole and triazine moieties in the tetrazole derived C_3N_5 materials.

The chemical structure of the intermediates was further demonstrated by NMR spectroscopy (**Figure S9**). The ^{15}N spectra of 3-AT and CN-200°C were tested with liquid NMR analysis (JNM-ECZ400R with 5mm Royal probe) dissolved by DMSO. The ^{13}C solid-state NMR spectra of all samples and ^{15}N solid-state NMR spectra of CN-300°C, CN-400°C, C_3N_5 (CN-500°C) and C_3N_4 were acquired on Bruker ADVANCE III 400 and 600 equipped with a 4 mm double resonance MAS NMR probe using the cross-polarization magic-angle spinning (CPMAS). ^{13}C spectra were referenced to TMS ($\delta(^{13}\text{C}) = 0.00$ ppm) by setting the high frequency ^{13}C peak of solid adamantane to 38.56 ppm. ^{15}N spectra were referenced to nitromethane $\delta(^{15}\text{N}) = 0.00$ ppm by setting the isotropic peak of a glycine sample (98 % ^{15}N) to -347.6 ppm.

The ^{13}C NMR spectra (**Figure S9a**) of 3-AT display two peaks at 157 and 148 ppm, attributed to N-C-NH (α) and $\text{N}_2\text{-C-NH}_2$ (β) of 3-AT, respectively, which was also proved by simulated ^{13}C NMR spectra (**Figure S10**). CN-200°C also exhibits two peaks at 163.5 and 144.1 ppm, attributed to two C atoms of 3-AT, respectively, which may be due to the fact that CN-200°C does not change the chemical structure of 3-AT because of the boiling point of 3-AT is 244.9°C, but high temperature heating for 3-AT affects the chemical shifts of the elements. Similarly, the ^{15}N NMR spectra of both 3-AT and CN-200°C display three peaks at -331, -188 and -170 ppm, attributed to C-NH₂ (N1), C-NH-N (N2) and C-N-C(N) (N3) in 3-AT, respectively (**Figure S9b**). The ^{13}C NMR spectra (**Figure S9a**) of CN-300°C display four peaks at 167, 163, 151 and 146 ppm, attributed to $\text{N}_2\text{-C-NH}_2$ (δ), $\text{NH}_x\text{-C-N}_2$ (β), CN_3 (γ) and $(\text{NH}_2)_2\text{-C-N}$ (α), respectively, which was also proved by simulated ^{13}C NMR spectra (**Figure S10**); while the ^{15}N NMR spectra (**Figure S9b**) of CN-300°C display seven peaks at -74.5, -133.8, -154.4, -166, -220.4, -237.6 and -261.1 ppm, attributed to N atoms in seven different chemical environments, respectively.

The ^{13}C NMR spectra of three samples (CN-400°C, C_3N_5 (CN-500°C) and C_3N_4) display two peaks at around 165 and 156 ppm and a weak peak at 169 ppm, attributed to $\text{C}_2\text{-NH}_x$ (α) and C_3N (β) in the heptazine units in C_3N_5 and C-NH-C=N (γ), respectively. Notably, a new peak at 111.9 ppm is observed for C_3N_5 , corresponding to $\text{C}_{\text{HN-C(H)=N}}$ (δ) in the triazole group, and all peaks of C_3N_5 is slightly shifted relative to C_3N_4 due to the triazole group in C_3N_5 . The ^{15}N NMR spectrum of C_3N_4 displays four signals at -156.6, -

191.6, -211.6 and -227.6 ppm, assigned to NC_2 (N4), central NC_3 (N3), bridged NH (N2) and NH_2 (N1), respectively [Angew. Chem. Int. Ed. 2020, 59, 16209-16217]. Similar to C_3N_4 , the ^{15}N NMR spectrum of CN-400°C and C_3N_5 (CN-500°C) exhibits similar peaks, and some new signals at -22.4, -146.7 and -288.8 ppm are observed. These new signals are attributed to C-N (N6)- N_2 , C=N (N5)-N and N-NH (N7) from the triazole group in C_3N_5 , respectively, due to the strong nitrogen-proton coupling with neighboring ammonia groups [Chem.–A Eur. J. 2012, 18, 16742-16753; J. Mater. Chem. A 2014, 2, 3200-3208]. Interestingly, almost no a signal of bridged NH (N2) was observed in the ^{15}N NMR spectrum of CN-400°C, which may be due to the formation of a single structural unit of C_3N_5 at 400°C. All these characterizations fully indicate that nitrogen-rich carbon nitride with triazole and two triazine groups in C_3N_5 was synthesized.

(Figure S9: in the revised Supplementary Information)

Figure S9: ^{13}C and ^{15}N NMR spectra (a and b) of 3-AT, CN-200°C, CN-300°C, CN-400°C, C_3N_5 (CN-500°C) and C_3N_4 and possible structural model representation (c). The ^{15}N spectra of 3-AT and CN-200°C were tested with liquid sample dissolved by DMSO, and the other samples were tested with solid samples.

(Figure S10: in the revised Supplementary Information)

Figure S10: Structural model and theoretical correspondence of ^{13}C NMR chemical shift produced from ChemDraw.

(Figure S11: in the revised Supplementary Information)

Figure S11: Micro Raman spectra of C_3N_4 and C_3N_5 . The circles and squares indicate Raman modes of triazine and triazole rings, respectively. 524nm shows the signal of Si, since the substrate is made of silicon.

The characteristic Raman peaks of g- C_3N_4 appear at 260, 316, 472, 715, 753, 980,

1233 and 1562 cm^{-1} . These peaks are assigned to breathing modes of the triazine ring [J. Phys. Chem. C 2011, 115, 7355-7363]. On the other hand, triazole-based C_3N_5 show additional vibration modes at 362, 618 and 667 cm^{-1} corresponding to breathing modes of the triazole ring [Angew. Chem. Int. Ed. 2018, 57, 17135-17140].

Based on the above results, we proposed the possible synthesis process. (Figure S12: in the revised Supplementary Information)

Figure S12: The possible synthetic steps of C_3N_5 .

Our modification to the manuscript: The corresponding description was added to the revised manuscript (Page 3) and Supplementary Information (Text S3, Figures 2c, 2d, S5 and S7-12 and Table S3).

2. The prepared material does present defects and a substantially narrower band gap which could be contributing to the enhanced photocatalytic performance. Claiming that such performance is higher than the state of the art, however, is quite misleading as the performance strongly depends also on reactor design so one could just make such claim if other catalysts have been tested under sonication and illumination with the same intensity (and the reactor design shown here is quite unconventional).

Author response: We appreciate the reviewer's professional comments. Your suggestion is very valuable. Indeed, the performance of the catalyst depends greatly on the reactor design. Here, we apply both ultrasonic and light for catalytic H_2O_2 production, and compared the performance of related reports presented in Figure S30 and Table S4. Since there are few reports on the production of H_2O_2 by applying both external force and light, I summarized the H_2O_2 production performance reported carbon nitride photocatalysts, all piezoelectric catalysts and all piezoelectric photocatalysts. Furthermore, we also added photocatalysts that previously lacked carbon nitride (Figure S30 and Table S4). However, it is very difficult to normalize the same intensity, especially for ultrasonic conversion efficiency, and most reports do not give the light intensity and frequency and power of ultrasound. Therefore, we have

further summarized the detailed reaction conditions including light intensity, ultrasound power and frequency, and rotational speed in all reference lists (**Table S4**) for further comparison. Overall, H₂O₂ yield we reported exceeds that of most catalysts.

(**Figure S30** and **Table S4**: in the revised Supplementary Information)

Figure S30: H₂O₂ production rates for C₃N₅ in this work compared with reported work.

Table S4. H₂O₂ production rates for C₃N₅ in this work compared with representative recently reported work.

Catalysts	Condition	Light intensity	Ultrasound conditions	sacrificial agent	H ₂ O ₂ (μmol g ⁻¹ h ⁻¹)	Reference
g-C ₃ N ₄ /PDI/rGO	Vis	-	-	-	24.1	32
Ag@U-g-C ₃ N ₄	Vis	100 mW cm ⁻²	-	-	70	33
Sb-SAPC15	Vis	400 mW cm ⁻²	-	-	91	34
DCN	Vis	-	-	20 vol% IPA	96.8	35
CTF-BDDBN	Vis	44.5 mW cm ⁻²	-	-	97	36
ZnPPc-NBCN	Vis	100 mW cm ⁻²	-	10 vol% IPA	114	21
Co ₁ /AQ/C ₃ N ₄	AM-1.5G	100 mW cm ⁻²	-	-	124	37
RF523	Vis	100 mW cm ⁻²	-	-	160	38
PEI/C ₃ N ₄	AM-1.5G	100 mW cm ⁻²	-	-	208.1	39
(K,P,O)-g-C ₃ N ₄	Vis	726.8 mW cm ⁻²	-	10 vol% EtOH	485.71	40

BP/CN	Vis	-	-	10 vol% IPA	540	41
CN ₄	Vis	-	-	10 vol% IPA	574	42
KPF ₆ /g-C ₃ N ₄	Vis	-	-	10 vol% EtOH	600	43
Nv-C≡N-CN	Vis	40 mW cm ⁻²	-	10 vol% IPA	3093	44
TP-PCN	Vis	-	-	10 vol% IPA	6530.8	45
ACNN	Vis	-	-	10 vol% IPA	10200	46
NOCN	Simulating sunlight	-	-	10 vol% IPA	11140	47
PCN-NaCA	Simulating sunlight	27 mW cm ⁻²	-	3.5 wt.% Glycerol	18700	48
CNF/SCNF-MS	Us/Stirring	-	45KW	-	62.8	49
BTO NSs	Us	-	180 W, 35 kHz	10 vol% MeOH	125.59	50
Au/BiVO ₄	Us	-	120 W, 40 kHz	4-CP	344.4	51
BiOCl	Us	-	-	Tris-buffered solution	420	52
SiO ₂ /PVDF-HFP	Us	-	300 W, 40 kHz	20 vol% EtOH	492	53
BiOCl	Us/Stirring	-	150 W, 53 kHz	-	560	54
C ₃ N _{5-x} -O	Us	-	-	-	615	55
C ₃ N ₄	Us/Stirring	-	150 W, 53 kHz/ 300rpm	-	680	56
UBTO-OV2	Us	-	300 W, 40 kHz	10 vol% EtOH	1611.2	57
LFZ	Us/Vis	-	180 W, 40 kHz	-	403	58
Bi ₄ NbO ₈ Br	Us/Vis	-	280 W, 40 kHz	10 vol% EtOH	792	59
g-C ₃ N ₄ /PDI-g-C ₃ N ₄	Us/Vis	-	200 W, 40 kHz	-	1040	60

Bulk-g-C ₃ N ₄	Us/Vis	-	240 W, 40 kHz	0.1 M glucose	1080	61
ZnS/In ₂ S ₃ /BTO	Us/Vis	100 mW cm ⁻²	150 W, 40 kHz	5M EtOH	1131.73	62
BaTiO ₃ :Nb/C	Us/Vis	100 mW cm ⁻²	150 W, 40 kHz	10 vol% EtOH	1360	63
BCVF	Us/Vis	-	100 W, 35 kHz	-	3173.53	64
C ₃ N ₅	Us/Vis	100 mW cm ⁻²	100 W, 40 kHz	10 vol% EtOH	1235.16	This work
C ₃ N ₅	Us/Simulating sunlight	115 mW cm ⁻²	100 W, 40 kHz	10 vol% EtOH	3809.52	This work

Our modification to the manuscript: The corresponding description was added to Supplementary Information (**Figure S30** and **Table S4**).

3. Additionally, the 2e⁻ selectivity has been assessed electrochemically by means of an RRDE which substantially different to a photocatalytic scenario, one depending on surface area and adsorption desorption and the other one depending more heavily in energy band position. Average quantum yield for instance would be more useful.

Author response: We appreciate the reviewer's professional comments.

AQE measurements: For AQE measurements, 20 mg of photocatalyst was dispersed in 40 mL of 10 vol% EtOH. A 300 W Xe-lamp with a band-pass filter of 380±15 nm, 420±15 nm, 450±15 nm, 500±15 nm, 550±15 nm, 600±15 nm or 650±15 nm was used as the incident light source. The light intensity was adjusted to be 4.52 mW cm⁻², 3.13 mW cm⁻², 3.64 mW cm⁻², 6.23 mW cm⁻², 7.07 mW cm⁻², 6.68 mW cm⁻² and 7.74 mW cm⁻², respectively. The irradiation area was controlled to be 4.9 cm². The amount of H₂O₂ production was analyzed after 1 h irradiation. AQE was calculated using the following equation:

$$\text{AQE}\% = 2 \times (N_{\text{H}_2\text{O}_2} \cdot N_A \cdot h \cdot c) / (I \cdot S \cdot t \cdot \lambda) \times 100\%$$

where $N_{\text{H}_2\text{O}_2}$ was the amount of H₂O₂ production (mol), N_A was the Avogadro constant ($6.022 \times 10^{23} \text{ mol}^{-1}$), h was the Planck constant ($6.626 \times 10^{-34} \text{ J}\cdot\text{s}$), c was the speed of light ($3 \times 10^8 \text{ m}\cdot\text{s}^{-1}$), I was the irradiation intensity ($\text{W}\cdot\text{cm}^{-2}$), S was the irradiation area (cm^2), t was the irradiation time (s) and λ was the wavelength of incident light (m).

The apparent quantum efficiency (AQE) of C₃N₅ was calculated at specific wavelengths and shown to approximately match with the UV-Vis spectrum (**Figure S25**). C₃N₅ exhibited an apparent quantum yield (AQY) of H₂O₂ production close to 7.3%, 6.7%, 4.1%, 2.1%, 1.4%, 1.2% and 0.9% at wavelengths of 380 nm, 420 nm, 450

nm, 500 nm, 550 nm, 600 nm and 650 nm, respectively. Overall, C_3N_5 exhibited the high photocatalytic activity of H_2O_2 production and highest uptake of O_2 , indicating the retarded charge recombination of dipole field in the triazole-based C_3N_5 backbone.

(**Figure S25**: in the revised Supplementary Information)

Figure S25: Apparent quantum yield (AQY) of H_2O_2 production at specific wavelengths superimposed with its UV-Vis absorption curve.

Our modification to the manuscript: The corresponding description was added to the revised manuscript (**Page 5**) and Supplementary Information (**Text S7** and **Figure S25**).

Summary Comments: Following this reasons i don't think this manuscript can be considered for publication in Nature Communications and i recommend the authors to either reword the discussion acknowledging the ambiguity of the structure, or try to elucidate further the structure by means of mass spectrometry or elemental analysis, which is very useful to prove the structure of melem analogues (RSC Adv. 2021, 11, 38862)

Author response: We highly appreciate the comments from the reviewer, and they are all considered in corrected manuscript. In the revised version, we further demonstrate our structure using organic elemental analysis, NMR, MALDI-TOF-MS, LC-TOF-MS, NEXAFS and Raman spectrometer.

Thank you very much again for your kind and appropriate comments. We are sure that these comments help improve the quality of our manuscript significantly.

Sincerely yours,

Mingshan Zhu

Reviewer #2 (Remarks to the Author):

Summary Comments: The authors have demonstrated the importance of the internal dipole field in nitrogen-rich carbon nitride photocatalysts and its impact on charge separation and H₂O₂ production. This work is of significant contribution to the field and deserves to be published in this journal after revisions addressing the following comments and questions:

Author response: We highly appreciate the comments from the reviewer, and they are all considered in corrected manuscript.

1. Clarify if the polarization induced by the dipole field is triggered by an external force, such as pressure or ultrasonication, or if it is a physical property of nitrogen-rich carbon nitride.

Author response: We appreciate the reviewer's critical comments. As the reviewer questioned, the formation of dipole fields is an inherent property of nitrogen-rich carbon nitride, which originates from the dipole induced by the material's own asymmetric structure. This dipole is defined as a pair of opposite charges " q " and " $-q$ " separated by a distance " d ". The direction of the dipole moment (p) in space is from negative charge " $-q$ " to positive charge " q ". This dipole moment with an electron cloud distribution forms a dipole field. It is worth noting that this dipole field can be adjusted by applying an external force such as pressure or ultrasonication to enhance the field strength, thereby promoting the separation efficiency of photogenerated carriers (**Figure 1b**). Here, polymerization of the triazole and triazine framework to form a nitrogen-rich carbon nitride (viz. C₃N₅) leads to asymmetry of the structure, and a dipole moment is generated by the noncoincidence of the positively and negatively charged centers with a value of 2.80 D for a single unit (**Figure 1d**). When the number of units increases to 6, the dipole moment is enhanced to 22.22 D (**Figure 1e**). This strong dipole moment means that dipole field-driven spontaneous polarization in C₃N₅ can be used to harness photogenerated charge separation kinetics.

(**Figures 1b, d and e:** in the revised manuscript)

Figure 1: b, Dipole moment and its electron cloud distribution, and dipole field and its change with external forces. **d**, Structural unit and dipole moments of C₃N₄ and C₃N₅ with positive and negative charge centers. **e**, Dipole moments of C₃N₄ and C₃N₅ with different structural unit numbers.

2. In Figure 3b, C₃N₄ produces H₂O₂ only with sonication despite lacking an internal dipole field. Recent reports suggest that C₃N₄ nanosheets exhibit in-plane piezoelectricity. Were the samples sonicated to form nanosheets before conducting photosynthesis?

Author response: We appreciate the reviewer's professional comments. As you mentioned, C₃N₄ can produce hydrogen peroxide under ultrasound without a dipole field, which can be due to following reasons: C₃N₄ has been shown to own in-plane piezoelectric response [J. Mater. Chem. A, 2019, 7, 20383; Adv. Mater. 2021, 33, 2101751], and its piezoelectric effect is due to the polarization field formed by the uneven distribution of positive and negative charge centers under external force, rather than its own structurally induced dipole field. Prior to photosynthesis, our samples were subjected by ultrasound treatment for 30 min to form nanosheets.

Our modification to the manuscript: The corresponding description was added to revised manuscript (**Page 8**).

3. Clarify if the H₂O₂ production in C₃N₅ under N₂ gas is due to oxygen pre-adsorbed on C₃N₅, and why the H₂O₂ production rate slightly decreases under air. Sonication of water can produce H₂O₂ through the sonolysis of H₂O molecule. Please carry out more control experiments to confirm how much of H₂O₂ is generated from the sonolysis of water alone.

Author response: We appreciate the reviewer's professional comments.

(1) Clarify if the H₂O₂ production in C₃N₅ under N₂ gas is due to oxygen pre-adsorbed on C₃N₅.

Based on your suggestion, we re-performed the N₂ atmosphere experiment, further checked the gas tightness, and extended the N₂ blast time to 20 min. The new experimental results demonstrated that almost no H₂O₂ was generated in the N₂ atmosphere (**Figure 3d**), and the results were roughly the same as the acoustic yield of water's sonication (**Figure S23**). These demonstrate that this may have been due to the poor gas tightness previously, which led to the entry of air and caused the elevated yield.

(2) why the H₂O₂ production rate slightly decreases under air.

As the following equation shows, the O₂ concentration is crucial for the synthesis of H₂O₂, and a higher O₂ concentration facilitates the positive proceeding of reaction and the synthesis of H₂O₂. Therefore, when a sealed reactor is filled with O₂, it can show a higher H₂O₂ yield compared to an open air atmosphere.

(3) Sonication of water can produce H_2O_2 through the sonolysis of H_2O molecule. Please carry out more control experiments to confirm how much of H_2O_2 is generated from the sonolysis of water alone.

Based on your suggestion, we conducted a direct ultrasonic experiment of pure water, and the result show that H_2O_2 of 21.3 μM was produced after 1h ultrasound.

(Figure 3d: in the revised manuscript)

Figure 3d: Effect of dissolved oxygen on H_2O_2 production for $\text{C}_3\text{N}_5/\text{Us}/\text{Vis}$ in 1 h.

(Figure S23: in the revised Supplementary Information)

Figure S23: a, Time profiles of H_2O_2 production via pure water with Us. b, Corresponding histograms of the H_2O_2 yield at 60 min.

Our modification to the manuscript: The corresponding description was added to revised manuscript (Page 5, Figure 3d) and Supplementary Information (Figure S23).

4. In Figure 4e, the peak strength is reduced after releasing pressure compared to before applying pressure. Clarify how the pressure affects the material's structure.

Author response: We appreciate the reviewer's professional comments. To monitor the charge migration process in photoexcited C_3N_5 under an external force, *in situ* pressure-dependent PL spectroscopy was performed. With gradually increasing pressure from ambient to 15 GPa, both C_3N_4 and C_3N_5 display similar changes from initial blue to green and finally to colorless at high pressure (Figures S48 and 49). These results are consistent with the chromaticity diagram of the Commission Internationale de l'Eclairage (CIE) (Figures S50). As shown in Figures S51 and 4e, as the pressure

increases, the PL peak positions of the two samples show an obvious redshift, and the peak intensity of C_3N_5 is significantly lower than that of C_3N_4 . When the pressure is released to 0 GPa, the peak position recovers, whereas the peak intensity decreases because the destruction of the structure under applied pressure weakens the luminescence of the material [Nanoscale 2020, 12, 12300-12307]. These results confirm that the inhibition efficiencies for charge recombination in photoexcited C_3N_4 and C_3N_5 increase with increasing external pressure, and the charge separation efficiency for C_3N_5 is higher than that for C_3N_4 . Therefore, the dipole field in C_3N_5 improves the charge migration behavior.

Recent reports have investigated the potential mechanism of these anomalous transitions in PL behavior via in situ high pressure XRD [Nanoscale 2020, 12, 12300-12307]. As the pressure increases, the layer stacking order of the carbon nitride material decreases while the layer interactions increase. In this case, the tri-s-triazine of carbon nitride shifting to the porous position of the nearest neighboring layer with an obvious drop in volume and the electron interactions are enhanced, especially, at the positions with larger electronic density where lone pair electrons of nitrogen occur in two samples under high pressure. This should further affect the PL emission related to the lone pair electrons of nitrogen. Hence, we observed that as two samples transforms into a less compressible state, and exhibit a more significant decrease in intensity under pressure.

Our modification to the manuscript: The corresponding description was added to Supplementary Information (**Note of Figure S51**).

5. Conduct scavenger tests with different concentrations to obtain more reliable quenching results.

Author response: We appreciate the reviewer's professional comments. Based on your suggestion, 1, 2 and 5 mM of scavengers were added to the initial solution in photocatalytic H_2O_2 production with Us. TBA for *OH have little effect on the production of H_2O_2 . It is clearly visible, p-BQ for $^*O_2^-$) mainly contribute to H_2O_2 production, and with the increase of p-BQ, the production of H_2O_2 gradually decreases.

(**Figure S56:** in the revised Supplementary Information)

Figure S56: Scavenger tests of C_3N_4 (a) and C_3N_5 (b) in photocatalytic H_2O_2 production with Us at 60 min (TBA for *OH , p-BQ for $^*O_2^-$, C=1, 2, 5 mM).

Our modification to the manuscript: The corresponding description was added to Supplementary Information (**Figure S56**).

6. Provide more details on the light-only experiment, such as whether it was conducted under stirring, and add a separate stirring experiment without light as a control. Also, clarify the differences between ultrasound and stirring.

Author response: We appreciate the reviewer's professional comments. Based on your suggestion, we added a separate stirring experiment without light as a control. In **Figure S24**, C_3N_4 and C_3N_5 produce almost no H_2O_2 with only stirring.

20 mg of catalyst was added to 40 mL of pure water containing ethanol (10 vol%) at pH=3. The catalyst was dispersed by ultrasonication for 10 min, and air was bubbled through the solution for 10 min. The reactor was kept at $25 \pm 0.5^\circ\text{C}$ with cooling circulating water and was irradiated at $\lambda \geq 420$ nm using a 300 W Xe lamp (PLS-SXE300D, Beijing Perfectlight Technology Co., Ltd) with a light intensity of 100 mW cm^{-2} , and simultaneously subjected to ultrasonication by an ultrasonic cleaner (40 kHz, 100 W, Jielimei, Kunshan, China). The light-only experiments were placed under a xenon lamp with stirring.

In catalytic reactions, stirring is often used for macroscopic mixing of liquids. However, it is difficult to achieve adequate mixing and generate enough force to excite the material for piezoelectric effects. While ultrasound provides microscopic mixing of liquid materials and induces an ultrasonic cavitation effect by applying a periodic force that causes the rupture of cavitation bubbles. This leads to a high pressure of up to 10^8 Pa at the non-homogeneous catalyst/water interface, which provides sufficient energy for electron excitation. Moreover, ultrasound can directly excite pure water through cavitation to produce H_2O_2 . These conclusions are also supported by the results presented in **Figure S23**.

(**Figure S24**: in the revised Supplementary Information)

Figure S24: Time profiles of H_2O_2 production by stirring-only with C_3N_4 and C_3N_5 .

Our modification to the manuscript: The corresponding description was added to revised manuscript (**Page 5**) and Supplementary Information (**Figure S24**).

7. Provide more details on how the authors conducted in-situ pressure-dependent PL spectra, such as how they achieved the high target pressure on quartz.

Author response: We appreciate the reviewer's critical comments. High-pressure experiments were performed using a symmetric diamond anvil cell (DAC). A pair of ultra-low fluorescence diamonds with anvil surface of 400 μm diameter was used to generate pressure for the *in situ* high-pressure PL experiments. A T301 stainless steel gasket pre-indented to the thickness of 45 μm was laser drilled to obtain a 150 μm diameter hole for loading the sample and a ruby ball. The ruby fluorescence technique was adopted for the pressure calibration in connection with the shift of the R1 line of the ruby fluorescence. The silicon oil was applied as pressure-transmitting medium around the sample. All the high-pressure experiments were conducted at room temperature. The high-pressure PL measurement were performed by using a combined home-made optical measurement system. The pressure-dependent PL spectra were measured by a semiconductor laser with an excitation wavelength of 355 nm, and recorded with an optical fiber spectrometer (Ocean Optics, QE65000). Note that the effects of different excitation laser intensities and luminous fluxes on the obtained PL intensity of carbon nitride were avoided by fixing all the parameters during each high-pressure PL experiment.

(Figure S61: in the revised Supplementary Information)

Figure S61: High-pressure experimental equipment DAC device diagram.

Our modification to the manuscript: The corresponding description was added to revised manuscript (Page 8-9) and Supplementary Information (Figure S61).

8. Evaluate the electron transport capacity of C_3N_5 and C_3N_4 using photocurrent and impedance measurements.

Author response: We appreciate the reviewer's professional comments. To reveal carriers migration of C_3N_5 , the transient piezoelectric current response is depicted in Figure S47, manifesting obvious and repeatable piezo-photo current signals with on/off of applying ultrasound or visible light. As shown in Figure S47, the current intensity (j) of C_3N_5 and C_3N_4 in various scenarios at 60 min follows the sequence $\text{C}_3\text{N}_5/\text{Us}/\text{Vis} > \text{C}_3\text{N}_4/\text{Us}/\text{Vis} > \text{C}_3\text{N}_5/\text{Vis} > \text{C}_3\text{N}_4/\text{Vis} > \text{C}_3\text{N}_5/\text{Us} > \text{C}_3\text{N}_4/\text{Us}$. The C_3N_5 under both ultrasound and visible light displays the highest migration rate of carriers, and

the result is matched well with H₂O₂ production. The electrochemical impedance spectroscopy (EIS) further explains a high-efficient carriers transfer of C₃N₅. The arc diameter of C₃N₅ is the smallest than that of C₃N₄. The smaller diameter is due to the lower charge transfer resistance and the faster mobility of electrons, indicating that the dipole field contributes to highest charges transfer efficiency of C₃N₅. These results fully demonstrate that the dipole field of triazole-based C₃N₅ can rapidly improve the separation rate of carriers.

(Figure S47: in the revised Supplementary Information)

Figure S47: a, Transient current density-time curves of C₃N₄ and C₃N₅ with on-off cycles of US, Vis, and US/Vis at a potential of -0.5 V in 0.5 M Na₂SO₄ solution, b, Nyquist plots of C₃N₄ and C₃N₅.

Our modification to the manuscript: The corresponding description was added to revised manuscript (Page 6) and Supplementary Information (Figure S47).

9. Include more references related to polarization-induced exciton dissociation in polymeric photocatalysts, such as Chem Catal 2, 1734-1747; Chem Catalysis 2 (7), 1517-1519.

Author response: We appreciate the reviewer's critical comments. Based on your suggestions, I have added corresponding references.

Our modification to the manuscript: The corresponding references has been added in the revised manuscript (Ref. 7 and 11).

Thank you very much again for your kind and appropriate comments. We are sure that these comments help improve the quality of our manuscript significantly.

Sincerely yours,

Mingshan Zhu

Reviewer #3 (Remarks to the Author):

Summary Comments: Dipole field in nitrogen-enriched carbon nitride Photosynthesis of H_2O_2 from O_2 and water is an ideal way to convert and store solar energy as well as the production of useful chemicals, because it can make full use of the photogenerated electrons and holes. Li et al. reported the photocatalytic H_2O_2 production on nitrogen-rich C_3N_5 . This present work has very fruitful characterizations and mechanism investigations. The structural engineering concept is novel and inspiring, and the results are rich and convincing. I think it can be published in Nature Communications after following issues being addressed.

Author response: We highly appreciate the positive comments from the reviewer, and they are all considered in corrected manuscript.

1. There are conceptual problems in Figure 1c. There always exists a surface band bending for a semiconductor, whose direction depends on the semiconductor type (n or p) and the exposed environment [Chem. Rev. 2012, 112, 5520-5551]. So the flat potential throughout the bulk and surface in (i) is not scientifically accurate. It is suggested to present a n-type case band bending, considering the n-type nature of the carbon nitride in this work. The concept demonstrated in (ii) is also problematic. Besides the band bending issue mentioned above, the transfer direction of charge carriers is not correct. As the energy level shown here generally represents the energy of electrons, the electrons should transfer from the high position to the low position of the energy level lines, and holes migrate to the opposite direction. In terms of the effect of dipole moment on the surface band bending, please refer to the paper [Angew. Chem. 2020, 132, 945-952].

Author response: We thank you for the reviewer's constructive comment. Your suggestion will have a great significance to improve the quality of my work. Based on your suggestion, we have reproposed an n-type case band bending model and the transfer direction of charge carriers [Chem. Rev. 2012, 112, 5520-5551]. It is worth noting that polar materials with dipole moments have a profound effect on surface energy band bending. The degree of upward band bending is more pronounced for polar materials accompanied by a dipole moment than for surfaces of nonpolar materials. Compared to nonpolar materials, the surfaces of polar materials with dipole moments exhibit more significant upward band bending [Angew. Chem. 2020, 132, 945-952; Chem. Soc. Rev. 2018, 47, 8238– 8262].

(Figure 1c: in the revised manuscript)

Our modification to the manuscript: The corresponding description has been added in the revised manuscript (**Page 2** and **Figure 1c**).

2. In Figure 2f, “an increase in the average surface potential” is used to characterize the effect of light on the surface potential. Yet it is not rational because as we can see, the difference between the lowest potential and the highest potential with light on C_3N_5 is about $940-890=50$ mV, even larger than the average increase of 30.61 mV. Actually, the CPD difference before and after light irradiation (Δ CPD) reflects the surface band bending extent [Chem. Soc. Rev. 2018, 47, 8238-8262]. It is suggested to present Δ CPD upon distance, and compare the two Δ CPD curves of C_3N_5 and C_3N_4 . Moreover, the huge variation of Δ CPD upon distance indicates the uneven spatial distribution of surface band bending, which may give more insights into photogenerated carriers’ kinetics if it can be correlated with the local structures of C_3N_5 and C_3N_4 .

Author response: We thank you for the reviewer’s constructive comment. Based on your valuable suggestions, we reselected the appropriate areas and presented the CPD according to the distance. To further identify surface charge modulation, the surface piezoelectric potential distribution with the surface morphologies of C_3N_5 and C_3N_4 was evaluated by Kelvin probe force microscopy (KPFM) (**Figures 2f** and **S15**). Upon illumination, we can observe that the KPFM images become brighter for n-type carbon nitride [Chem. Soc. Rev. 2018, 47, 8238– 8262]. The results agree well with the CPD increases for n-type semiconductors. The range of the contact potential difference (CPD) for C_3N_5 under dark condition is approximately 859~888 mV, which is apparently higher than the range of 607~635 mV for C_3N_4 ; that for C_3N_5 under light condition is approximately 908~946 mV, which is also higher than the range of 620~649 mV for C_3N_4 . Moreover, C_3N_5 with light irradiation exhibits an increase in the average surface potential of ≈ 44.42 mV relative to that without light, while C_3N_4 with light shows an increase of ≈ 10.41 mV (**Figure 2f**). This is because the spontaneous polarization induced by the dipole field of C_3N_5 amplifies the directional charge transfer upon light irradiation. As previously reported, compared to nonpolar materials, the surfaces of polar materials with dipole moments exhibit more significant upward band bending [Chem. Soc. Rev. 2018, 47, 8238–8262], which effectively inhibits charge recombination upon irradiation with modulated light, the overall surface potential of the whole material is increased (**Figure S15**). Furthermore, the huge variation of Δ CPD upon distance indicates the uneven spatial distribution of surface band bending, which may be correlated with the local polarization structures of C_3N_5 and C_3N_4 .

Your suggestion will have a great significance to improve the quality of my work. It is a valuable and far-reaching proposal. However, due to time constraints, we will conduct study on your suggestion in the follow-up work, especially the role of relationship between Δ CPD and local structures of material. I truly believe that your opinion will significantly improve the quality of our future work.

(Figures 2f: in the revised manuscript)

Figure 2f: Surface potential from KPFM images of C_3N_4 and C_3N_5 with (w/) and without (w/o) light irradiation.

(Figures S15: in the revised Supplementary Information)

Figure S15: Surface morphologies and corresponding KPFM potential images of C_3N_4 (a) and C_3N_5 (b). **i**, AFM 3D topography images. **ii**, AFM 2D topography images. **iii**, **iv**, contact potential difference (CPD) of C_3N_4/C_3N_5 with and without light. The arrow is the selected surface potential area.

Our modification to the manuscript: The corresponding description has been added in the revised manuscript (Page 4, Figure 2f) and Supplementary Information (Figure S15).

3. Solid state NMR is the most efficient and direct proof to investigate the detailed structure of carbon nitride-based materials. The author also used this technology to confirm the molecular structure. However, the present resolution of ^{15}N NMR spectra is not convincing enough to get the conclusion. More precise characterizations should be performed to clearly demonstrate the structure.

Author response: We appreciate the reviewer’s critical comments. In the revised version, we further demonstrate our structure using organic elemental analysis (UNICUBE-Elementar), Nuclear magnetic resonance (NMR, Bruker ADVANCE III 400 and 600 for solid-state NMR spectra and JNM-ECZ400R for liquid-state NMR spectra), Matrix-assisted laser desorption/ionization–time of flight mass spectrometry (MALDI-TOF-MS, Bruker Autoflex Speed TOF/TOF), liquid chromatography time-of-flight mass spectrometer (LC-TOF-MS, AB Sciex Triple TOF 5600) and near-edge X-ray absorption finestructure (NEXAFS) at the beamline BL14W1 station of the Beijing Synchrotron Radiation Facility.

Here, carbon nitride structures at different temperatures including 200, 300 and 400°C using 3-AT (viz. CN-200°C, CN-300°C and CN-400°C) were also synthesized for comparison. First, we evaluate the elemental ratio of C to N for all samples using organic elemental analysis. The CHN results of C₃N₅ (CN-500°C) confirm that the average weight percentages of C and N are 33.1% and 63.1%, respectively (**Table S3**). The average atomic rate of C/N is thus determined to be 0.611, which is very close to the theoretical value (0.60), confirming the successful synthesis of C₃N₅. Specifically, for 3-AT and CN-200°C, the elemental proportion of CNH has barely changed, which may be due to the fact that the boiling point of 3-AT is 244.9°C, implying that the chemical composition has not changed. For CN-300°C, CN-400°C and C₃N₅ (CN-500°C), with the increase of synthesis temperature, the percentage weight of N and H decreases gradually, this may be due to the possible release of ammonia and hydrogen during the synthesis process. The total mass fraction of 3-AT exceeded 100%, due to the water absorption of the sample.

(**Table S3:** in the revised Supplementary Information)

Table S3. Elemental composition of 3-AT, CN-200°C, CN-300°C, CN-400°C and C₃N₅ (CN-500°C) as determined by organic elemental analysis (C, N, H).

Sample	Nitrogen (wt%)	Carbon (wt%)	Hydrogen (wt%)	Carbon/ Nitrogen (Atomic ratio)
3-AT	66.8	28.8	6.76	0.503
	66.8	28.7	6.29	0.501
	66.7	28.7	5.54	0.502
CN-200°C	66.8	28.9	5.05	0.505
	66.5	28.7	5.77	0.504
	66.1	28.8	5.33	0.508
CN-300°C	64.8	29.1	3.95	0.524
	64.8	29.2	4.86	0.526
	64.7	29.3	4.84	0.528
CN-400°C	63.9	31.9	4.4	0.582
	63.6	31.7	4.79	0.581
	63.7	31.8	4.08	0.582
C ₃ N ₅ (CN-500°C)	63.1	33.1	3.21	0.611
	63.1	33.2	3.34	0.613
	63.0	33.0	3.42	0.611

(Figure S7: in the revised Supplementary Information)

Figure S7. MALDI-TOF-MS spectra (a) of CN-400°C; Major species (b) constituting the precipitate formed from CN-400°C. The solid sample was adopted for testing.

MALDI-TOF-MS was used to determine the chemical structure of intermediates. **Figure S7a** presents the whole MALDI-TOF-MS spectra of the intermediate of CN prepared at 400°C. Since the amino groups on the carbon nitride are easily charged with positive protons, we tested the intermediates in positive ion mode. The peaks with m/z of 64, 71, 96, 106, 127, 192, 196 and 211 Da are observed in the MALDI-TOF MS spectra of CN-400°C. Therefore, some possible molecular structures of the intermediates in the CN polymerization are listed in **Figure S7b**. Based on this, the above m/z can correspond to the eight positively ionized products of the four molecules as above. This implies the possible existence of these four intermediates. Other excess peak may be due to the effect of incomplete polymerization of 3-AT and other impurities.

(Figure S8: in the revised Supplementary Information)

Figure S8. LC-TOF-MS spectra (a) of H_2O and CN-400°C; Major species (b) constituting the precipitate formed from CN-400°C. DMSO was used as a solvent for CN-400°C, and DMSO was used as a control sample.

LC-TOF-MS was used to determine the chemical structure of intermediates. **Figure S8a** presents the whole LC-TOF-MS spectra of the intermediate of CN prepared at 400°C. We also tested the intermediates in positive ion mode, because the amino groups on the carbon nitride are easily charged with positive protons. The peaks with m/z of 43, 64, 71, 96, 106, 127, 192, 196 and 211 Da are observed in the LC-TOF-MS spectra of CN-400°C. Therefore, the positively ionization mode of these possible intermediates is shown in are listed in **Figure S8b**. These results are also very consistent with the results of MALDI-TOF-MS. This also demonstrates the possible existence of these four intermediates.

(Figure S5: in the revised Supplementary Information)

Figure S5. C K-edge NEXAFS spectra of C₃N₄ and C₃N₅.

(Figure 2c: in the revised manuscript)

Figure 2c. N K-edge NEXAFS spectra of C₃N₄ and C₃N₅.

Near-edge X-ray absorption finestructure (NEXAFS) analysis was also used to explore the chemical bonds of triazole-based C₃N₅. The C K-edge NEXAFS spectra (Figure S5) of C₃N₅ and C₃N₄ show characteristic excitations including $1s \rightarrow \pi^*_{\text{out of plane C=C}}$ at ~ 285.2 eV and $1s \rightarrow \pi^*_{N-C=N}$ at 288.0 eV [Nat. Commun. 2014, 5, 3783]. Compared with C₃N₄, C₃N₅ displays a blue shift of $1s \rightarrow \pi^*_{N-C=N}$ excitation, which is attributed to increased N-C=N bond strength upon the formation of triazole moiety. The N K-edge NEXAFS spectra (Figure 2c) of two samples show similar response at 399.3 and 402.5 eV, attributing to the $1s \rightarrow \pi^*_{N-C=N}$ and π^*_{C-N} resonance. Note that, relative to C₃N₄, triazole-based C₃N₅ displays a red shift of $1s \rightarrow \pi^*_{N-C=N}$ excitation and a new peak at 401.1 eV,

which are attributed to the formation of triazole moiety and $1s \rightarrow \pi^*$ heterocyclic N-N of triazole, respectively [Nat. Commun. 2014, 5, 3783]. These experimental findings provide strong evidence of the coexistence of triazole and triazine moieties in the tetrazole derived C_3N_5 materials.

The chemical structure of the intermediates was further demonstrated by NMR spectroscopy (**Figure S9**). The ^{15}N spectra of 3-AT and CN-200°C were tested with liquid NMR analysis (JNM-ECZ400R with 5mm Royal probe) dissolved by DMSO. The ^{13}C solid-state NMR spectra of all samples and ^{15}N solid-state NMR spectra of CN-300°C, CN-400°C, C_3N_5 (CN-500°C) and C_3N_4 were acquired on Bruker ADVANCE III 400 and 600 equipped with a 4 mm double resonance MAS NMR probe using the cross-polarization magic-angle spinning (CPMAS). ^{13}C spectra were referenced to TMS ($\delta(^{13}C) = 0.00$ ppm) by setting the high frequency ^{13}C peak of solid adamantane to 38.56 ppm. ^{15}N spectra were referenced to nitromethane $\delta(^{15}N) = 0.00$ ppm by setting the isotropic peak of a glycine sample (98 % ^{15}N) to -347.6 ppm.

The ^{13}C NMR spectra (**Figure S9a**) of 3-AT display two peaks at 157 and 148 ppm, attributed to N-C-NH (α) and N_2 -C-NH₂ (β) of 3-AT, respectively, which was also proved by simulated ^{13}C NMR spectra (**Figure S10**). CN-200°C also exhibits two peaks at 163.5 and 144.1 ppm, attributed to two C atoms of 3-AT, respectively, which may be due to the fact that CN-200°C does not change the chemical structure of 3-AT because of the boiling point of 3-AT is 244.9°C, but high temperature heating for 3-AT affects the chemical shifts of the elements. Similarly, the ^{15}N NMR spectra of both 3-AT and CN-200°C display three peaks at -331, -188 and -170 ppm, attributed to C-NH₂ (N1), C-NH-N (N2) and C-N-C(N) (N3) in 3-AT, respectively (**Figure S9b**). The ^{13}C NMR spectra (**Figure S9a**) of CN-300°C display four peaks at 167, 163, 151 and 146 ppm, attributed to N_2 -C-NH₂ (δ), NH_x-C-N₂ (β), CN₃ (γ) and (NH₂)₂-C-N (α), respectively, which was also proved by simulated ^{13}C NMR spectra (**Figure S10**); while the ^{15}N NMR spectra (**Figure S9b**) of CN-300°C display seven peaks at -74.5, -133.8, -154.4, -166, -220.4, -237.6 and -261.1 ppm, attributed to N atoms in seven different chemical environments, respectively.

The ^{13}C NMR spectra of three samples (CN-400°C, C_3N_5 (CN-500°C) and C_3N_4) display two peaks at around 165 and 156 ppm and a weak peak at 169 ppm, attributed to C₂N-NH_x (α) and C₃N (β) in the heptazine units in C_3N_5 and C-NH-C=N (γ), respectively. Notably, a new peak at 111.9 ppm is observed for C_3N_5 , corresponding to C_{HN-C(H)=N} (δ) in the triazole group, and all peaks of C_3N_5 is slightly shifted relative to C_3N_4 due to the triazole group in C_3N_5 . The ^{15}N NMR spectrum of C_3N_4 displays four signals at -156.6, -191.6, -211.6 and -227.6 ppm, assigned to NC₂ (N4), central NC₃ (N3), bridged NH (N2) and NH₂ (N1), respectively [Angew. Chem. Int. Ed. 2020, 59, 16209-16217]. Similar to C_3N_4 , the ^{15}N NMR spectrum of CN-400°C and C_3N_5 (CN-500°C) exhibits similar peaks, and some new signals at -22.4, -146.7 and -288.8 ppm are observed. These new signals are attributed to C-N (N6)-N₂, C=N (N5)-N and N-NH (N7) from the triazole group in C_3N_5 , respectively, due to the strong nitrogen-proton coupling with neighboring ammonia groups [Chem.-A Eur. J. 2012, 18, 16742-16753; J. Mater. Chem. A 2014, 2, 3200-3208]. Interestingly, almost no a signal of bridged NH (N2) was observed in the ^{15}N NMR spectrum of CN-400°C, which may be due to the formation of a single

structural unit of C_3N_5 at 400°C . All these characterizations fully indicate that nitrogen-rich carbon nitride with triazole and two triazine groups in C_3N_5 was synthesized.

(Figure S9: in the revised Supplementary Information)

Figure S9. ^{13}C and ^{15}N NMR spectra (a and b) of 3-AT, CN- 200°C , CN- 300°C , CN- 400°C , C_3N_5 (CN- 500°C) and C_3N_4 and possible structural model representation (c). The ^{15}N spectra of 3-AT and CN- 200°C were tested with liquid sample dissolved by DMSO, and the other samples were tested with solid samples.

(Figure S10: in the revised Supplementary Information)

Figure S10. Structural model and theoretical correspondence of ^{13}C NMR chemical shift produced from ChemDraw.

Our modification to the manuscript: The corresponding description was added to the revised manuscript (Page 3) and Supplementary Information (Text S3, Figures 2c, 2d, S5 and S7-10 and Table S3).

4. In presence of alcohols as electron donors, some other carbon nitride-based catalysts show higher efficiency in H₂O₂ production, such as ACS Catal. 2020, 10, 24, 14380–14389, Nat. Commun., 2021, 12, 3701, Angew. Chem. Int. Ed., 2021, 60(48): 25546-25550, Chem Catal., 2022, 2(7): 1720-1733 etc. The Figure S20 should also include these activities in the latest publications.

Author response: We appreciate the reviewer’s professional comments. We added the corresponding references in Figure S30 and Table S4.

(Figure S30 and Table S4: in the revised Supplementary Information)

Figure S30: H₂O₂ production rates for C₃N₅ in this work compared with reported work.

Table S4. H₂O₂ production rates for C₃N₅ in this work compared with representative recently reported work.

Catalysts	Condition	Light intensity	Ultrasound conditions	sacrificial agent	H ₂ O ₂ (μmol g ⁻¹ h ⁻¹)	Reference
g-C ₃ N ₄ /PDI/rGO	Vis	-	-	-	24.1	32
Ag@U-g-C ₃ N ₄	Vis	100 mW cm ⁻²	-	-	70	33
Sb-SAPC15	Vis	400 mW cm ⁻²	-	-	91	34
DCN	Vis	-	-	20 vol% IPA	96.8	35
CTF-BDBBN	Vis	44.5 mW cm ⁻²	-	-	97	36

ZnPPc- NBCN	Vis	100 mW cm ⁻²	-	10 vol% IPA	114	21
Co ₁ /AQ/C ₃ N ₄	AM-1.5G	100 mW cm ⁻²	-	-	124	37
RF523	Vis	100 mW cm ⁻²	-	-	160	38
PEI/C ₃ N ₄	AM-1.5G	100 mW cm ⁻²	-	-	208.1	39
(K,P,O)-g- C ₃ N ₄	Vis	726.8 mW cm ⁻²	-	10 vol% EtOH	485.71	40
BP/CN	Vis	-	-	10 vol% IPA	540	41
CN ₄	Vis	-	-	10 vol% IPA	574	42
KPF ₆ /g- C ₃ N ₄	Vis	-	-	10 vol% EtOH	600	43
Nv-C≡N- CN	Vis	40 mW cm ⁻²	-	10 vol% IPA	3093	44
TP-PCN	Vis	-	-	10 vol% IPA	6530.8	45
ACNN	Vis	-	-	10 vol% IPA	10200	46
NOCN	Simulating sunlight	-	-	10 vol% IPA	11140	47
PCN-NaCA	Simulating sunlight	27 mW cm ⁻²	-	3.5 wt.% Glycerol	18700	48
CNF/SCNF- MS	Us/Stirring	-	45KW	-	62.8	49
BTO NSs	Us	-	180 W, 35 kHz	10 vol% MeOH	125.59	50
Au/BiVO ₄	Us	-	120 W, 40 kHz	4-CP	344.4	51
BiOCl	Us	-	-	Tris- buffered solution	420	52
SiO ₂ /PVDF -HFP	Us	-	300 W, 40 kHz	20 vol% EtOH	492	53
BiOCl	Us/Stirring	-	150 W, 53 kHz	-	560	54
C ₃ N _{5-x} -O	Us	-	-	-	615	55
C ₃ N ₄	Us/Stirring	-	150 W, 53 kHz/ 300rpm	-	680	56
UBTO-OV2	Us	-	300 W, 40 kHz	10 vol% EtOH	1611.2	57

LFZ	Us/Vis	-	180 W, 40 kHz	-	403	58
Bi ₄ NbO ₈ Br	Us/Vis	-	280 W, 40 kHz	10 vol% EtOH	792	59
g- C ₃ N ₄ /PDI- g-C ₃ N ₄	Us/Vis	-	200 W, 40 kHz	-	1040	60
Bulk-g- C ₃ N ₄	Us/Vis	-	240 W, 40 kHz	0.1 M glucose	1080	61
ZnS/In ₂ S ₃ / BTO	Us/Vis	100 mW cm ⁻²	150 W, 40 kHz	5M EtOH	1131.73	62
BaTiO ₃ :Nb /C	Us/Vis	100 mW cm ⁻²	150 W, 40 kHz	10 vol% EtOH	1360	63
BCVF	Us/Vis	-	100 W, 35 kHz	-	3173.53	64
C ₃ N ₅	Us/Vis	100 mW cm ⁻²	100 W, 40 kHz	10 vol% EtOH	1235.16	This work
C ₃ N ₅	Us/Simula ting sunlight	115 mW cm ⁻²	100 W, 40 kHz	10 vol% EtOH	3809.52	This work

Our modification to the manuscript: The corresponding description has been added in the revised Supplementary Information (**Figure S30** and **Table S4**).

5. To evaluate the efficiency of the catalyst, activity under monochromatic light should be performed to calculate the apparent quantum yield.

Author response: We appreciate the reviewer's professional comments.

AQE measurements: For AQE measurements, 20 mg of photocatalyst was dispersed in 40 mL of 10 vol% EtOH. A 300 W Xe-lamp with a band-pass filter of 380±15 nm, 420±15 nm, 450±15 nm, 500±15 nm, 550±15 nm, 600±15 nm or 650±15 nm was used as the incident light source. The light intensity was adjusted to be 4.52 mW cm⁻², 3.13 mW cm⁻², 3.64 mW cm⁻², 6.23 mW cm⁻², 7.07 mW cm⁻², 6.68 mW cm⁻² and 7.74 mW cm⁻², respectively. The irradiation area was controlled to be 4.9 cm². The amount of H₂O₂ production was analyzed after 1 h irradiation. AQE was calculated using the following equation:

$$\text{AQE\%} = 2 \times (N_{\text{H}_2\text{O}_2} \cdot N_A \cdot h \cdot c) / (I \cdot S \cdot t \cdot \lambda) \times 100\%$$

where $N_{\text{H}_2\text{O}_2}$ was the amount of H₂O₂ production (mol), N_A was the Avogadro constant ($6.022 \times 10^{23} \text{ mol}^{-1}$), h was the Planck constant ($6.626 \times 10^{-34} \text{ J}\cdot\text{s}$), c was the speed of light ($3 \times 10^8 \text{ m}\cdot\text{s}^{-1}$), I was the irradiation intensity ($\text{W}\cdot\text{cm}^{-2}$), S was the irradiation area (cm^2), t was the irradiation time (s) and λ was the wavelength of

incident light (m).

The apparent quantum efficiency (AQE) of C_3N_5 was calculated at specific wavelengths and shown to approximately match with the UV-Vis spectrum (**Figure S25**). C_3N_5 exhibited an apparent quantum yield (AQY) of H_2O_2 production close to 7.3%, 6.7%, 4.1%, 2.1%, 1.4%, 1.2% and 0.9% at wavelengths of 380 nm, 420 nm, 450 nm, 500 nm, 550 nm, 600 nm and 650 nm, respectively. Overall, C_3N_5 exhibited the high photocatalytic activity of H_2O_2 production and highest uptake of O_2 , indicating the retarded charge recombination of dipole field in the triazole-based C_3N_5 backbone.

(**Figure S25**: in the revised Supplementary Information)

Figure S25: Apparent quantum yield (AQY) of H_2O_2 production at specific wavelengths superimposed with its UV-Vis absorption curve.

Our modification to the manuscript: The corresponding description was added to the revised manuscript (**Page 5**) and Supplementary Information (**Text S7** and **Figure S25**).

6. The *in-situ* pressure-dependent UV-vis absorption spectra characterization is very impressive. It may be inspiring to future works on the pressure-induced band gap modulation. It is suggested to provide a detailed picture of the set-up for this measurement to make it clearer to the readers.

Author response: We appreciate the reviewer's professional comments. We added a detailed picture of the set-up for *in situ* high pressure UV-vis absorption spectroscopy system.

(Figure S38: in the revised Supplementary Information)

Figure S38: Photograph of in situ high pressure UV absorption spectroscopy system.

Our modification to the manuscript: The corresponding description was added in Supplementary Information (**Figure S38**).

7. Minor problems: The pictures for adsorption configuration in Figure 6a is fuzzy. Clear and distinct pictures should be provided. “The surface plane O_2 is adsorbed in a Pauling-type manner at the N active sites in C_3N_5 ” should be “On the surface plane, O_2 is adsorbed ...”

Author response: We appreciate the reviewer’s careful comments. New clear pictures and sentences have been updated.

Our modification to the manuscript: The corresponding description was updated in the revised manuscript (**Figure 6a** and **Page 1-2**).

Thank you very much again for your kind and appropriate comments. We are sure that these comments help improve the quality of our manuscript significantly.

Sincerely yours,

Mingshan Zhu

REVIEWERS' COMMENTS

Reviewer #1 (Remarks to the Author):

I believe the authors have done a very thorough job revising the manuscript, have provided experimental data that supports the C₃N₅ structure (such as elemental analysis, MS, NMR or XAS) and have set a more thorough comparison of the piezo-photocatalytic performance with the state of the art. In this state, the manuscript can be considered for publication in Nature communications.

Reviewer #2 (Remarks to the Author):

The revised manuscript addressed the raised issues properly and thoroughly. I have no further comments.

Reviewer #3 (Remarks to the Author):

Authors have addressed my comments well. I feel it can be accepted for publication now.